# Doa10/MARCH6 architecture interconnects E3 ligase activity with lipid-binding transmembrane channel to regulate SQLE

J. Josephine Botsch [1,2], Roswitha Junker [1], Michèle Sorgenfrei [3], Patricia P. Ogger [4], Luca Stier[1,2], Susanne von Gronau[1], Peter J. Murray [4], Markus A. Seeger [3], Brenda A. Schulman [1] ✉ & Bastian Bräuning [1] ✉

Transmembrane E3 ligases play crucial roles in homeostasis. Much protein and organelle quality control, and metabolic regulation, are determined by ER-resident MARCH6 E3 ligases, including Doa10 in yeast. Here, we present Doa10/MARCH6 structural analysis by cryo-EM and AlphaFold predictions, and a structure-based mutagenesis campaign. The majority of Doa10/MARCH6 adopts a unique circular structure within the membrane. This channel is established by a lipid-binding scaffold, and gated by a flexible helical bundle. The ubiquitylation active site is positioned over the channel by connections between the cytosolic E3 ligase RING domain and the membrane-spanning scaffold and gate. Here, by assaying 95 MARCH6 variants for effects on stability of the well-characterized substrate SQLE, which regulates cholesterol levels, we reveal crucial roles of the gated channel and RING domain consistent with AlphaFold-models of substrate-engaged and ubiquitylation complexes. SQLE degradation further depends on connections between the channel and RING domain, and lipid binding sites, revealing how interconnected Doa10/MARCH6 elements could orchestrate metabolic signals, substrate binding, and E3 ligase activity.

Molecular machineries within the ER are central to membrane protein homeostasis. Here new membrane and secretory proteins are matured and misfolded or unnecessary proteins are identified and directed towards degradation through ER-associated protein degradation (ERAD)[1–8]. During this process, ERAD substrates are directed to E3 ubiquitin ligases and subsequently retrotranslocated for proteasomal degradation. In yeast, three primary E3 ubiquitin ligases execute ERAD: the Hrd1 complex, processing luminal and membrane proteins[9–13]; the Asi complex acting on the inner nuclear membrane[14,15]; and Doa10, a prominent ER-resident ligase that ubiquitylates soluble and membrane proteins[16–21].

Doa10 is categorized within the RING class of E3 ubiquitin ligases. In E1-E2-E3 cascades, RING E3s engage with ubiquitin-loaded E2 enzymes, and facilitate the transfer of ubiquitin from E2 to the E3-bound substrate[22,23]. Doa10, initially identified through a genetic screen for factors regulating the mating factor Matα2, has various substrates including Erg1[24], Pgc1[25], Sbh2[26], and Ubc6[27]. Notably, Ubc6 not only acts as a substrate but also serves as Doa10's cognate E2 enzyme. Doa10 also collaborates with another E2 enzyme, Ubc7, to lengthen polyubiquitin chains on its substrates[16,22,28], leading to their extraction from the ER and subsequent degradation, a process that involves the Cdc48 ATPase[19].

[1]Department of Molecular Machines and Signaling, Max Planck Institute of Biochemistry, Am Klopferspitz 18, 82152 Martinsried, Germany. [2]Technical University of Munich, School of Natural Sciences, Munich, Germany. [3]Institute of Medical Microbiology, University of Zurich, Gloriastrasse 28/30, 8006 Zurich, Switzerland. [4]Research Group of Immunoregulation, Max Planck Institute of Biochemistry, Am Klopferspitz 18, 82152 Martinsried, Germany. ✉e-mail: schulman@biochem.mpg.de; bastianwangbraeuning@gmail.com

The human ortholog of Doa10, MARCH6 (or alternatively MARCHF6/TEB4)[29], shares conserved features with its yeast counterpart. Several pathways moderated by MARCH6 align with those of Doa10, especially concerning protein ubiquitylation in sterol biosynthesis[24,30,31]. Moreover, the human E2 enzyme Ube2J2, orthologous to yeast's Ubc6, works in tandem with MARCH6, affecting proteins including SQLE[32–34]. Recent findings expand MARCH6's role to several other metabolic pathways[34–39]. The importance of MARCH6's role in metabolic regulation is underscored by sterols stimulating ubiquitin-dependent degradation of SQLE[31].

Despite progress in understanding Doa10 and MARCH6 functions, structure-guided functional analysis surveying the entire conserved portion of these E3 ligases remains absent. E3 ligases are typically characterized using mutagenesis combined with functional assays[40,41]. This approach, however, faces challenges with transmembrane protein complexes due to the necessity of co-reconstitution[19]. And while budding yeast offers tools for studying E3 ligases[42,43], Doa10 presents unique hurdles, notably its gene's toxicity to *E. coli*[16,19,44]. Here we address these challenges using a cell-based method, measuring a fluorescent SQLE reporter, to analyze 95 MARCH6 variants selected for study based on cryo-EM data and AlphaFold (AF)[45] predictions. Our findings reveal key structural elements and emphasize a multi-site mechanism governing transmembrane E3 ligases.

## Results

### A fluorescence-based reporter assay to track MARCH6 activity in human cells

To functionally map MARCH6, we devised a cell-based assay employing a fluorescent reporter: SQLE, a known MARCH6 substrate, C-terminally fused to mCherry[24,30,46] (Fig. 1a). For the purpose of analyzing MARCH6-dependent degradation, we used a K562-dCas9-zim3 cell line[47], incorporating a dual-fluorophore reporter construct (SQLE-mCherry-P2A-sfGFP). Given the cotranslational production of SQLE-mCherry and GFP, the mCherry:GFP ratio offers a metric for the posttranslational stability of the SQLE-mCherry fusion, measurable via flow cytometry[48,49]. Subsequently, we used a lentiviral dual-sgRNA vector[47] to target MARCH6, thereby depleting the native ligase through dCas9-zim3-driven CRISPRi. We then reintroduced either wild-type (WT) or mutant MARCH6 and selected the expressing cells post-lentiviral transduction with blasticidin.

Our methodology underwent several validation steps. We verified the reduction of MARCH6 mRNA in CRISPRi-treated cells using quantitative PCR (Supplementary Fig. 1a). As direct MARCH6 detection is not feasible due to the lack of available antibodies[34], we monitored its activity by assessing endogenous SQLE levels post-MARCH6 depletion (Supplementary Fig. 1b), which, as expected, rose compared to a nontargeting (N.T.) guide. Moreover, WT MARCH6 reintroduction postdepletion resulted in normal SQLE levels (Supplementary Fig. 1c), a function not achieved by a MARCH6 variant missing the RING domain. We then showed that the reintroduced MARCH6[WT] successfully facilitated degradation of the fluorescent SQLE reporter as monitored by flow cytometry (Supplementary Fig. 1d). Importantly, a C-terminal FLAG-tag on the reintroduced MARCH6 did not interfere with its activity (Supplementary Fig. 1c, d). As a further verification, we assessed the SQLE reporter's reflection of native MARCH6 function by applying cholesterol (as Chl-CD) to reporter cell lines. Consistent with prior findings[31], rising cholesterol levels triggered faster MARCH6-driven loss of SQLE reporter signal (Fig. 1b). Upon MARCH6 depletion, our fluorescent SQLE reporter revealed notable stabilization without added Chl-CD and an absence of Chl-CD-driven degradation. Finally, contrasting SQLE levels in cells with reintroduced MARCH6[WT], MARCH6[ΔRING], or a known ligase-defective variant (MARCH6[G885L])[43], displayed varied reporter stabilities, confirming the assay's robust dynamic range (Fig. 1c). Collectively, these outcomes validate the SQLE

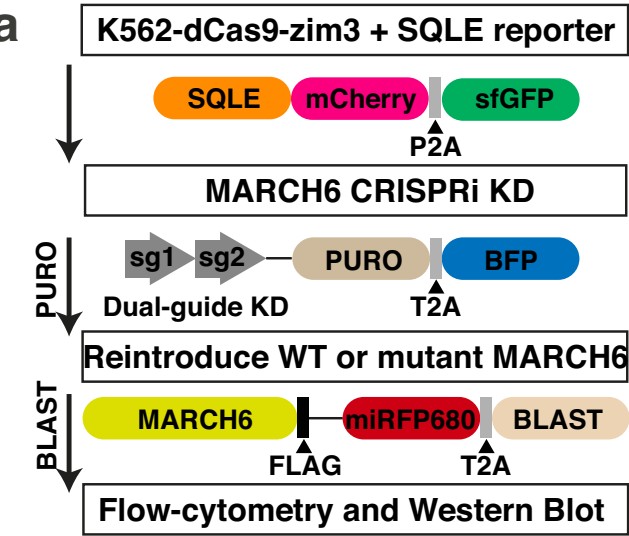

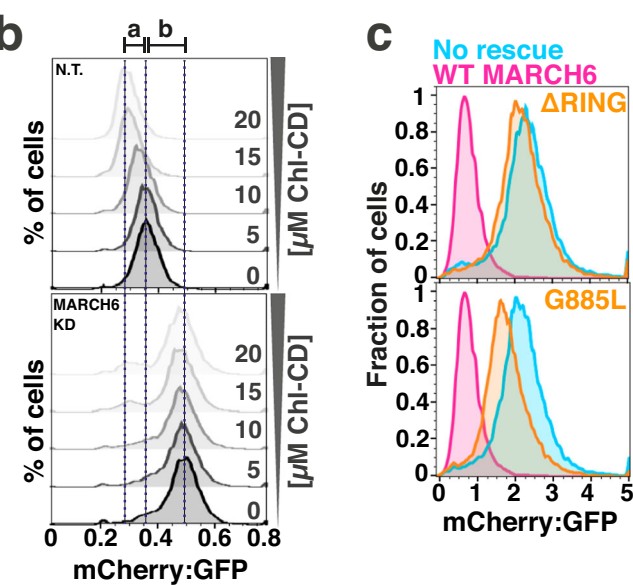

**Fig. 1 | Overview of the fluorescent SQLE reporter system to track MARCH6 activity in cells. a** Experimental design for depleting MARCH6 and assessing reintroduced ligase mutants' ability to complement SQLE reporter degradation. The SQLE reporter has a C-terminal mCherry fusion, separated by a P2A sequence from downstream sfGFP. It is stably introduced into K562-dCas9-zim3 CRISPRi cells via lentivirus and sorted as GFP-positive. MARCH6-depleted reporter cells are created using two sgRNAs targeting MARCH6 in a puromycin-resistant lentiviral vector. BFP fluorophore allows virus titering and gating in flow cytometry. WT or mutant MARCH6 is reintroduced via a FLAG-tagged lentiviral vector, generating a homogenous MARCH6-expressing population through blasticidin treatment. miRFP680 enables virus titering and gating in flow cytometry. **b** SQLE reporter response to cholesterol levels depends on MARCH6. K562-dCas9-zim3 cells with SQLE reporter and stably expressing either N.T. or MARCH6 KD sgRNA were treated with increasing Chl-CD levels for 8 h, then analyzed by flow cytometry. a: mCherry:GFP ratio difference between 0 μM and 20 μM Chl-CD in N.T. cells; b: mCherry:GFP ratio difference between N.T. and MARCH6 KD cells without Chl-CD. Results representative of three independent biological replicates. **c** SQLE reporter's dynamic range for detecting MARCH6 mutant defects. K562-dCas9-zim3 cells with SQLE reporter and stable MARCH6-targeting sgRNA expression were transfected with MARCH6[WT], MARCH6[ΔRING], miRFP680 only (no rescue), or intermediate-defect mutant MARCH6[G885L]. Post-blasticidin selection, flow cytometry analysis gated on miRFP680-positive cells. Histogram presentation of relative mCherry fluorescence normalized to GFP as an expression control. Results representative of four biological replicates.

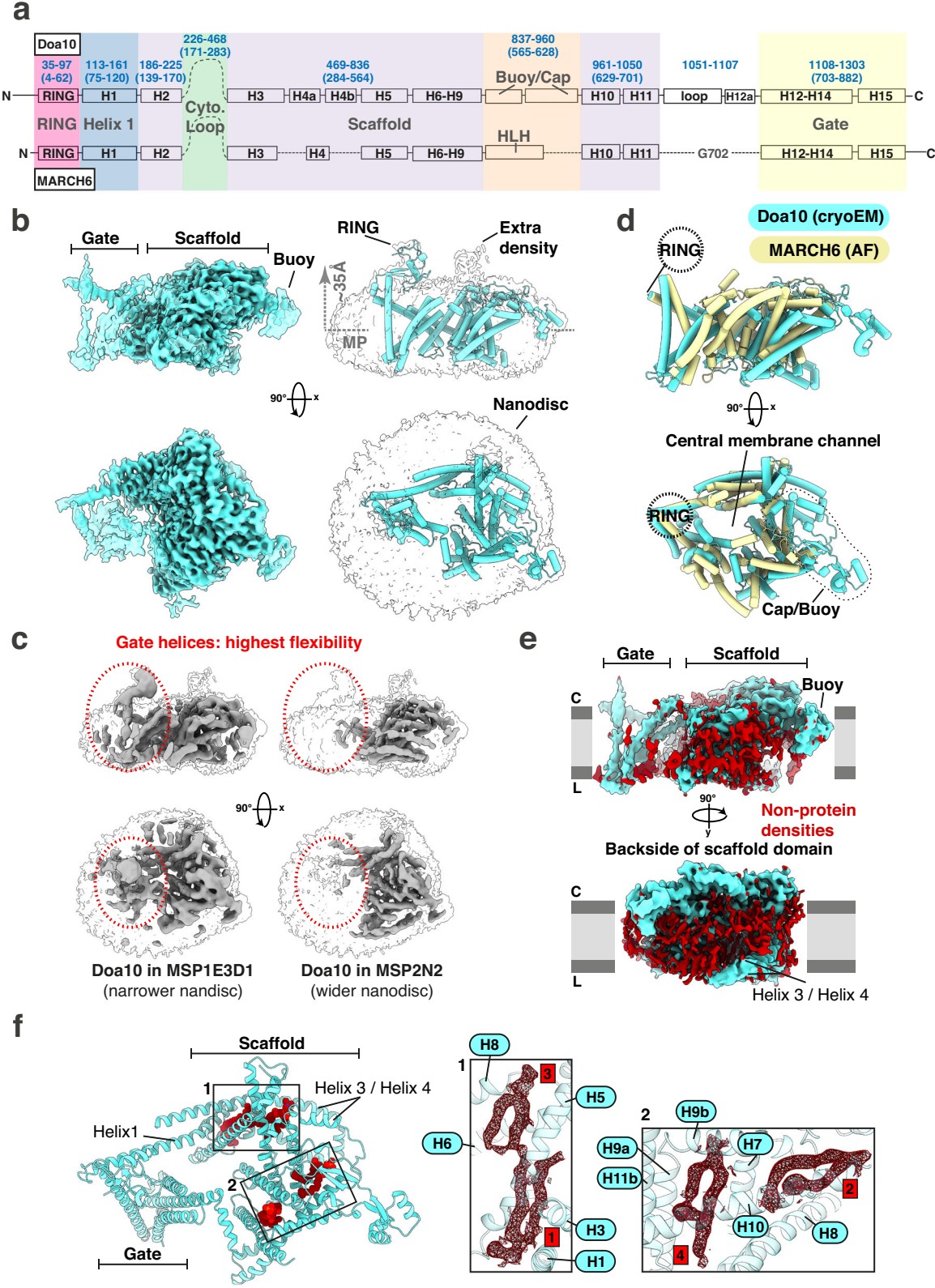

reporter's capability to sensitively track MARCH6 activity and facilitate extensive mutagenesis studies on E3 ligase function.

**Cryo-EM and structure predictions of MARCH6-family E3 ligases**

Before our mutagenesis campaign, we wanted to analyze MARCH6-family ligases structurally. We used two strategies for designing structure-based mutations. First, we looked at AF[45] models for both

human MARCH6 and yeast Doa10. Second, we determined the Doa10 structure with cryo-EM (Fig. 2). We expressed Doa10 in insect cells with its E2 Ubc6. After tandem affinity purification and size exclusion chromatography, the protein complex was reconstituted in MSP1E3D1 lipid nanodiscs with yeast total lipid extract (Supplementary Fig. 2a). To help structure determination, a sybody[50,51] against Doa10-Ubc6 was bound to the complex in our final cryo-EM sample, yielding a Doa10

**Fig. 2 | Cryo-EM analysis of the Doa10-Ubc6 complex reconstituted in lipid nanodiscs. a** Domain map and comparison of Doa10 (top) and MARCH6 (bottom). Residue numbering (blue, Doa10: top, MARCH6: brackets), transmembrane-spanning helices (H1-H15) and domains (colored boxes) are shown for both proteins. Most domains are conserved between Doa10 and MARCH6 including the RING domain, scaffold (with Helix 1) and C-terminal gate helices. The largest differences are localized to a region capping the scaffold domain (Buoy and Cap domain in Doa10, Helix-Loop-Helix (HLH) segment in MARCH6). We were unable to build the long cytoplasmic region between Helix 2 and 3. **b** High-resolution cryo-EM reconstructions. *Left*: Side and top view of the high-resolution cryo-EM map of the Doa10-Ubc6 complex at two different overlaid thresholds. The flexible gate and buoy domain are only being visible at lower threshold. *Right*: Side and top view of the modeled Doa10 structure as cartoon representation within the high-resolution cryo-EM at a very low threshold to see the nanodisc boundaries and any low-resolution density. Dotted grey arrow denotes distance of RING domain to membrane mid-plane (MP). **c** Side and top view of the low-resolution cryo-EM maps of the Doa10 complex in MSP1E3D1 (left side, ∅ ~ 12.9 nm) or MSP2N2 (right side, ∅ 15-16.5 nm). The silhouette of the high-resolution cryo-EM map of the Doa10 complex at very low threshold is depicted for both maps for better comparison of the flexible parts (red circle). **d** Comparison of the Doa10 cryo-EM structure (light blue) and the predicted AlphaFold model of human MARCH6 (light yellow). **e** Non-protein densities surrounding the Doa10 cryo-EM map. Density corresponding to the modeled Doa10 structure is colored in light blue, while unmodeled density is depicted in dark red. The unmodeled density most likely belongs to lipids surrounding the Doa10 complex within the lipid nanodisc. **f** Tightly bound lipids in the Doa10 cryo-EM model. Four bound lipids are depicted as spheres and colored in dark red. A close-up of the areas with tightly bound lipids shows the density for those lipids as dark red mesh. Doa10 helices in close proximity to the lipids are numbered.

reconstruction at 3.58 Å resolution (Supplementary Fig. 2b, c, Supplementary Table 1). The cryo-EM map is dominated by a well-resolved, large domain of nanodisc-enwrapped transmembrane helices (Supplementary Fig. 3a). Although our cryo-EM sample contained Ubc6 and the sybody, we did not observe unambiguous density for either, even after extensive 3D classification (Supplementary Fig. 2c). Efforts to obtain cryo-EM structures of human MARCH6, alone or with Ube2J2, in different reconstitution systems, remained unsuccessful.

The cryo-EM map allowed for the identification of all predicted Doa10 transmembrane helices (Fig. 2a, Supplementary Fig. 4a, b). Side chains were visible for a horseshoe-shaped region we call the scaffold domain (residues 113–1050) (Fig. 2a, b, Supplementary Figs. 3a, 4b). Inside this domain, two long transmembrane helices (Helix 1: residues 113–161 and Helix 3: residues 469–524) pack nearly perpendicular to the rest of the membrane scaffold comprised of Helix 2 and Helices 4–11 (Fig. 2a, Supplementary Fig. 4a). The cytosolic region between residues 240–468, predicted to be unstructured (Fig. 2a), was not visible in the map. Examining the map at lower threshold showed a bundle of four transmembrane helices (Helices 12–15, residues 1108–1303) (Fig. 2a). This area, which we call the flexible gate, had significant differences in the complex reconstituted in narrow and wide nanodiscs, seen in our medium-resolution cryo-EM data (Fig. 2c, Supplementary Fig. 5a, b), indicating potential for this part of the protein to adopt different orientations. Flexibility of this region is also suggested by the poorer local map resolution of gate helices compared to the scaffold (Supplementary Fig. 3a). The scaffold and flexible gate domains together form a wide central transmembrane channel. The AF RING domain model was docked into the map at lower thresholds, hovering about ~35 Å above the lipid bilayer mid-plane (Fig. 2b, Supplementary Fig. 3b). In summary, the cryo-EM data showed Doa10 has a central transmembrane channel formed by a rigid scaffold and a flexible gate with the cytosolic RING domain positioned above.

Comparing our Doa10 cryo-EM structure with the Doa10 AF model shows that the structure is predicted with high confidence (RMSD: 1.2 Å), validating the use of AF for predicting the fold. Comparing the Doa10 structure to the MARCH6 AF prediction (RMSD: 1.1 Å for the scaffold) unveils the conservation of the structural organization, the transmembrane helices' circular layout, and the RING's relative placement from yeast to human (Fig. 2d). Differences are the relative shortening between Helices 3 and 5 in MARCH6 towards the scaffold's backside, opposite the central membrane channel and RING domain, and between Helices 9 and 10 (Fig. 2a, d). In Doa10, this latter area forms the small adjoining cap (residues 890–960) and buoy (residues 837–888) domains. In MARCH6, it's a predicted helix-loop-helix structure (residues 564–628). In Doa10, the flexible gate connects to the scaffold through a short helical cytosolic segment (residues 1051–1107). In MARCH6, the gate and scaffold are close, centered on G702.

The cryo-EM map showed many non-protein densities around the Doa10 transmembrane domain (Fig. 2e). From their shapes and our reconstitution of Doa10 in yeast lipid nanodiscs, we presume these densities are weakly-associated lipids. Also, some phospholipids with clear cryo-EM densities are firmly intercalated within the scaffold domain (Fig. 2f, Supplementary Fig. 4c).

## A structure-guided mutagenesis campaign reveals MARCH6 sites important for SQLE degradation

By employing structural analyses based on both AF prediction and cryo-EM data, we identified a set of 95 mutants (some with multiple residue substitutions, thus 195 mutations surveyed in total) across MARCH6 to discover functionally-important regions (mutant effects and statistical analysis are listed in the Supplementary Data 3). Testing these for SQLE reporter stability showed diverse effects, which are discussed in the following sections. As a quality control readout, we also examined protein abundance of each mutant by anti-FLAG immunoblotting, alongside MARCH6$^{WT}$ and MARCH6$^{ΔRING}$ as benchmarks (Supplementary Fig. 6a). This analysis revealed that only a few mutants were less abundant compared to MARCH6$^{WT}$, suggesting that the protein fold was retained in the majority of variants studied. Additional qPCR analysis of a subset of lower-expressing mutants confirmed that mRNA levels were comparable to WT levels. Thus, we surmise that lower protein abundance in some mutants could result from post-translational instability (Supplementary Fig. 6b). In line with MARCH6 auto-regulation through auto-ubiquitylation and subsequent degradation[29,31], we observed accumulation of several mutants similar to MARCH6$^{ΔRING}$, likely reflecting severe defects in catalyzing ubiquitylation.

By normalizing median mCherry:GFP ratios to the two controls, MARCH6$^{WT}$ (0% defective) and MARCH6$^{ΔRING}$ (100% defective), we could visualize mutant effects either mapped onto the AF predicted three-dimensional structure of MARCH6 (Fig. 3a), or as a bar chart for each mutant sorted into different categories (Fig. 3b). Interestingly, we also discovered mutations causing even more SQLE destabilization than MARCH6$^{WT}$, with MARCH6$^{Q866A/Q869A}$ being the most potent gain-of-function (GOF) mutant. We confirmed that the MARCH6 mutant effects measured using our exogenous SQLE reporter reflect their effects on endogenous SQLE for a subset of these MARCH6 rescue cell lines by immunoblot analysis (Supplementary Fig. 7a). When probing the same cell lines for endogenous levels of the lipid droplet protein PLIN2—another protein whose amount is regulated by MARCH6, although potentially not as a direct ubiquitylation target[38,39]—we observed similar PLIN2 stabilization in some of the more robust mutants but weaker or even opposite effects in other positions (Supplementary Fig. 7b).

Mutations displaying consistent defects or GOF across biological replicates fall within regions of the predicted MARCH6 structure (Fig. 3b) that we broadly categorize as: 1) within or proximal to the

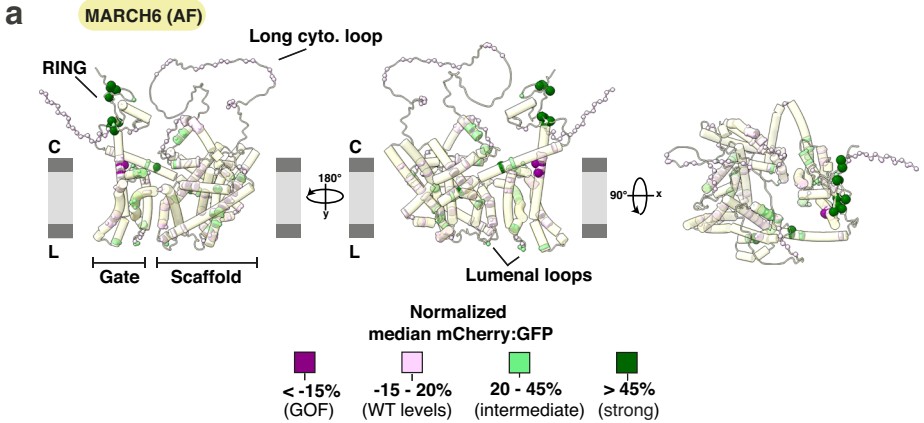

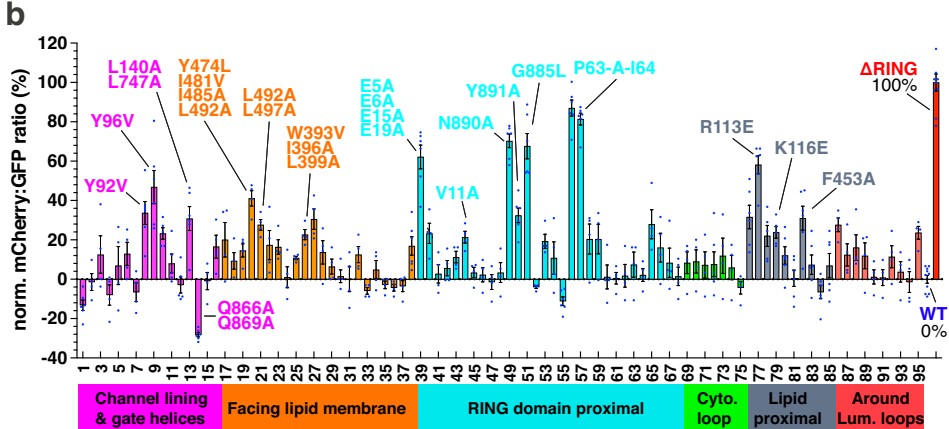

**Fig. 3 | A broad mutagenesis campaign across MARCH6 uncovers functionally important regions for SQLE degradation. a** Normalized median mCherry:GFP ratios for each MARCH6 mutant mapped onto AF prediction in fluorescent SQLE reporter cells. The predicted model displays targeted residues as spheres, colored by normalized mCherry:GFP ratio (MARCH6$^{\Delta RING}$ as 100% defective, MARCH6$^{WT}$ as 0% defective). Severe defects are colored dark green, intermediate defects are shown in light green, light rose indicates levels comparable to WT, and purple marks GOF mutants. The ER lipid bilayer is schematically shown in grey (C = cytosol, L = ER lumen). **b** Bar graph showing normalized median mCherry:GFP ratios for each MARCH6 variant in SQLE reporter cells. Ratios normalized to MARCH6$^{WT}$ (0%) and MARCH6$^{\Delta RING}$ (100%). Bars are colored by mutant categories discussed in the main text. The X-axis shows mutant numbering, which is detailed in the Supplementary Data 3. Mean, standard error of the mean (SEM) as error bars and individual data points for each mutant are shown. Statistical analysis including mean of the mCherry:GFP ratio, SEM, number of replicates and *P* values of the ANOVA pairwise comparison of the median mCherry:GFP ratios of each mutant to WT are listed in the Supplementary Data 3. Mutants discussed in the text are labeled. Source data are provided as a Source Data file.

cytosolic RING domain; 2) lining the exterior hydrophobic surface facing the lipid environment, which are potential binding sites for interaction partners including Ube2J2; 3) lining the interior of the central membrane channel enclosed by scaffold and gate domains; 4) in the MARCH6 scaffold domain corresponding to positions where we observed tightly bound phospholipids in the Doa10 cryo-EM map (Fig. 2f); and 5) within and surrounding luminal loops. We did not reproducibly observe strong defects for mutations in a conserved long cytosolic loop (Figs. 2a and 3).

**Mutants near the RING suggest the catalytic importance of positioning above the central membrane channel**

Several defective MARCH6 mutants center around the RING domain (Fig. 3). Many exhibit an increase in MARCH6 protein levels (Supplementary Fig. 6a). The defects in controlling SQLE reporter levels, combined with mutant accumulation presumably resulting from reduced auto-ubiquitylation, indicate a reduced capacity to catalyze ubiquitylation.

For structural insights into mutants suggested to play a catalytic role, we used AF-multimer[52] to model a MARCH6-ubiquitin-Ube2J2 complex for both the full-length protein complex, spanning the membrane (Fig. 4a), and only the soluble catalytic domains focusing

on the ubiquitylation active site (Supplementary Fig. 8a). The model resembled known RING-E2-ubiquitin structures (Supplementary Fig. 8a) in the active conformation[53,54]. We thus tested the predicted positioning of the catalytic moieties using an ubiquitin discharge assay. This assay follows the release of ubiquitin from the thioester bond with the catalytic domain from Ube2J2, stimulated by interactions between E2-ubiquitin and the isolated RING domain. Indeed, adding WT MARCH6 RING domain to the ubiquitin-loaded Ube2J2 catalytic domain accelerates ubiquitin discharge in our in vitro assay (Fig. 4b and Supplementary Fig. 8b).

To further assess the positioning of the catalytic domains, we mutated conserved residues predicted to be localized at the interfaces between these domains (Supplementary Fig. 8a). First, we mutated MARCH6 RING V11 predicted to interact with a hydrophobic patch in UBE2J2 (Supplementary Fig. 8a). Consistent with the model, mutations to aspartic acid or alanine lead to a strong or intermediate defect, respectively, in RING domain-mediated ubiquitin discharge from the catalytic Ube2J2-ubiquitin complex (Fig. 4b, Supplementary Fig. 8b). Next, we targeted the predicted interface between the E2 catalytic domain with the ubiquitin I44 patch (Fig. 4c, Supplementary Fig. 8a, c, d). Introducing bulky residues at this interface (S120F, S120R, T116F, T116R) results in impaired ubiquitin discharge with WT ubiquitin.

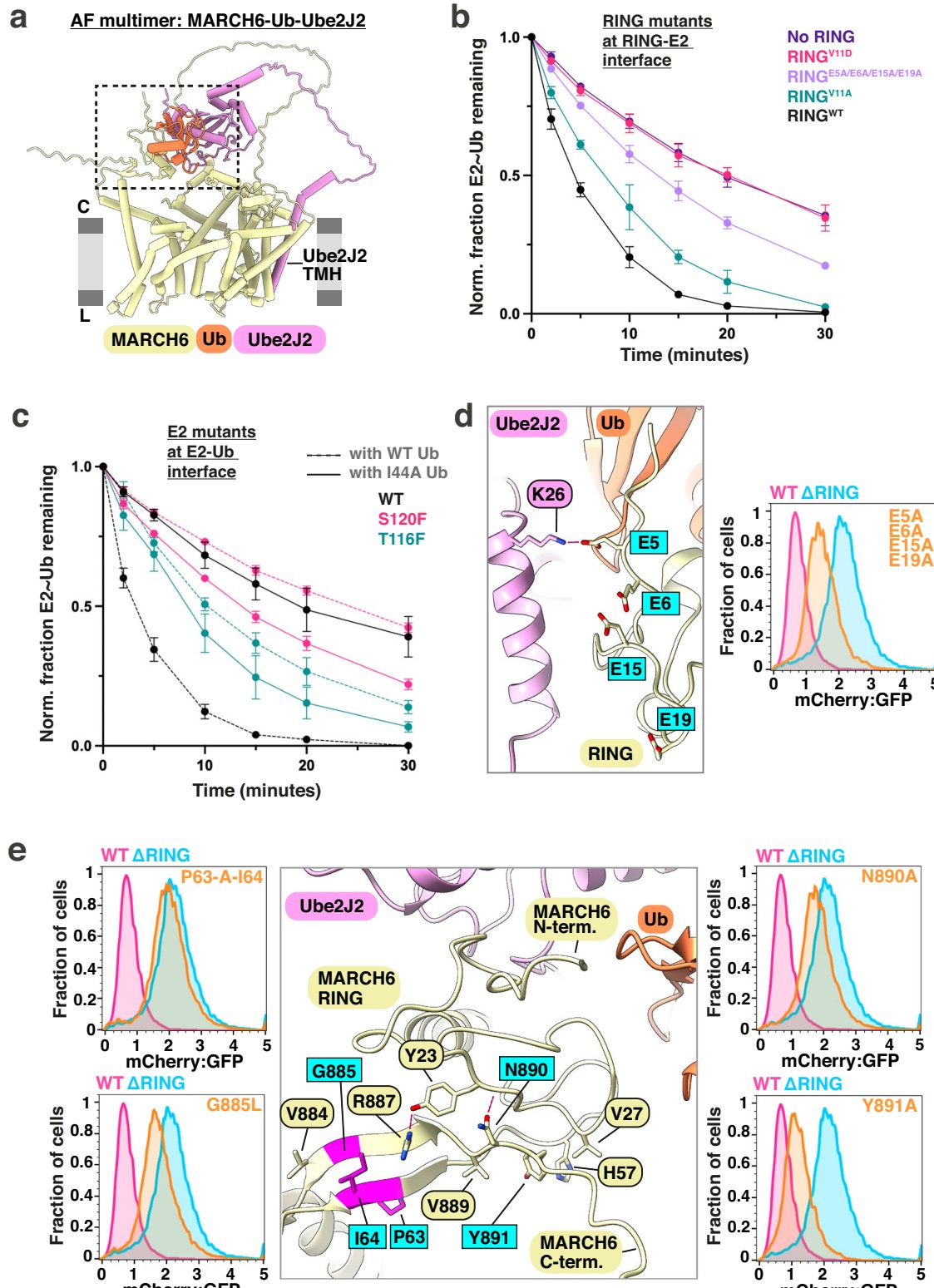

These discharge defects are partially rescued with a ubiquitin I44A mutation designed to compensate for the larger opposing E2 residues (Fig. 4c, Supplementary Fig. 8c). Furthermore, E2 mutations to glycine have milder effects, which are not rescued by ubiquitin I44A in agreement with the model (Supplementary Fig. 8c).

In our broad mutagenesis campaign, we found mutations in the MARCH6 N-terminus (MARCH6$^{E5A/E6A/E15A/E19A}$) highly defective in mediating SQLE degradation (Fig. 3b, cyan). Our MARCH6-Ube2J2-ubiquitin

AF model predicts MARCH6 E5 to form a bond with Ube2J2 residue K26 (Fig. 4d), suggesting additional catalytic domain stabilization by this N-terminus. Indeed, the isolated MARCH6$^{E5A/E6A/E15A/E19A}$ RING domain has a defect in stimulating ubiquitin discharge from Ube2J2 compared to WT (Fig. 4b), supporting a role of the N-terminus. Finally, we also observed accumulation of MARCH6$^{E5A/E6A/E15A/E19A}$ protein, emphasizing the importance of the MARCH6 N-terminus for catalysis, including auto-degradation (Supplementary Fig. 6a).

**Fig. 4 | Positioning of the MARCH6 RING domain above a membrane channel by N- and C-terminal elements. a** AF multimer model of the ternary MARCH6-ubiquitin(Ub)-UBE2J2 complex. Membrane boundaries are shown in grey, the UBE2J2 TMH is emphasized and the catalytic moieties are encircled by a black box. **b** Discharge of the loaded UBE2J2 catalytic domain with ubiquitin (UBE2J2-Ub) is stimulated by the isolated MARCH6 RING domain in vitro. The normalized fraction of remaining UBE2J2-Ub is shown for seven time points (0–30 min). Discharge for WT RING domain, no RING domain and three RING domain mutants (V11A, V11D, E5A/E6A/E15A/E19A) are depicted. Error bars present the SEM. Three independent experiments were conducted. Source data is provided in Supplementary Fig. 8b and Source Data file. **c** Mutants at the interface of UBE2J2 and ubiquitin are important for ubiquitin discharge in vitro. Comparison of WT UBE2J2 and two mutants (T116F and S120F) with either WT ubiquitin (dashed line) or I44A ubiquitin (solid line). The discharge is stimulated with WT MARCH6 RING domain for all samples. The normalized fraction of UBE2J2-Ub is depicted with error bars representing the SEM of the data. Three independent experiments were conducted.

Source data is provided in Supplementary Fig. 8d and Source Data file. **d** MARCH6 acidic N-terminus is crucial for SQLE degradation. *Left*: Close-up of predicted AF model of MARCH6 N-terminus, with mutated residues highlighted in cyan. The predicted hydrogen bond between MARCH6 E5 and UBE2J2 K26 is shown as a red dotted line. *Right*: Flow cytometry panel comparing SQLE reporter levels in cells re-expressing MARCH6^WT, MARCH6^ΔRING, or MARCH6^E5A/E6A/E15A/E19A mutant after MARCH6 depletion. Histogram depiction of normalized mCherry:GFP fluorescence. Representative result shown from six independent biological replicates. **e** Pillar-like structure formed by a beta-sheet downstream of the RING domain with the C-terminus orients MARCH6 RING domain functionally. *Middle*: Close-up of MARCH6 RING domain interactions with selected C-terminal elements. Predicted hydrogen bonds from AF multimer are shown as magenta dashed lines. Targeted residues within the beta-sheet are emphasized in magenta and labeled in cyan. *Left/Right*: Flow cytometry panels of targeted residues (cyan). Representative result shown from six independent biological replicates.

The sequence following the RING domain forms a β-sheet with the MARCH6 C-terminus (Fig. 4e). This β-sheet establishes a pillar-like structure positioning the RING domain above the central membrane channel in our Doa10 cryo-EM map and the AF models. Previous research examining effects of Doa10 and MARCH6 mutations mapping to the pillar demonstrated functional importance[43,55]. Accordingly, in our screen, alanine insertions in this β-sheet (MARCH6^P63-A-I64, MARCH6^T62-A-P63) resulted in nearly complete SQLE reporter degradation loss and an extent of mutant MARCH6 accumulation comparable to MARCH6^ΔRING (Supplementary Fig. 6a).

The MARCH6 AF structure shows pillar residue interactions with the RING domain. A prior study[43] revealed the MARCH6^N890A mutant's strong SQLE degradation defect. Predicted interactions between the pillar and RING include R887 from the β-sheet contacting Y23, and Y891 within a groove created by V27, H57, A60, and V889 (Fig. 4e). The structural model suggests that the MARCH6^G885L mutant within the β-sheet, could clash with sidechains of I64 and V884 or lead to β-sheet distortion. We found MARCH6^G885L defective in SQLE reporter degradation, similar to MARCH6^Y891A. Together, our findings suggest that these MARCH6 N- and C-terminal regions anchor the RING domain above the central membrane channel to recruit E2-ubiquitin for substrate ubiquitylation.

In the the predicted full-length MARCH6-Ube2j2-ubiquitin structure, a linker connects the Ube2j2 catalytic domain to a transmembrane helix (TMH). The Ube2j2 TMH fits into a groove on the MARCH6 scaffold backside (Supplementary Fig. 9a). Several of our screen mutants on the MARCH6 exterior interface with this Ube2j2 TMH (Supplementary Fig. 9b). Some of these mutants (MARCH6^Y474L/I481V/I485A/L492A, MARCH6^W393V/I396A/L399A, and MARCH6^L492A/L497A) exhibit intermediate defects in SQLE degradation while being expressed at levels comparable to MARCH6^WT. The AF-predicted yeast Doa10-Ubc6-ubiquitin complex also has the E2 C-terminal TMH binding a groove on the E3 scaffold backside (Supplementary Fig. 9c). A recent study showed the cap/buoy domains bind Ubc6 elements near the E2 TMH[55]. The groove's conservation from yeast to human and the AF prediction make a strong case for E2 TMH binding to Doa10/MARCH6, but mutations in this area impact SQLE, not MARCH6 stability, raising the possibility of alternate functions besides ubiquitylation.

## The MARCH6 membrane channel is functionally important and the predicted site of SQLE engagement
The most striking feature of the structural data is the positioning of the E3 ligase RING domain—and a modeled ubiquitylation active site—hovering above the transmembrane channel established by the scaffold and gate domains. Conservation of this general structural organization between Doa10 and MARCH6 (Fig. 2d), raises the questions as to what the channel properties are, and if the channel could play important roles in E3 ligase function. In our experimentally

determined Doa10 structure, this channel has an approximate diameter of 15 Å, measured from gate Helix 13 to scaffold Helix 7 (Fig. 5a). For perspective, this is large enough to surround one TMH with bulky amino acids and slightly larger than the translocon channel formed by the Sec61 complex in its wider state[56,57]. In both orthologs, scaffold elements Helix 1, Helix 2, Helix 7, and Helix 9 line one side of the channel (Supplementary Fig. 10a). However, due to different orientations between the gate helix bundle and the scaffold, distinct gate helices expand the membrane channel in MARCH6 compared to Doa10.

Three observations suggest the membrane channel dimensions are influenced by the location of gate helices relative to the scaffold domain. First, our cryo-EM reconstructions of Doa10 in two nanodisc sizes differed significantly in the gate region, with density almost entirely absent in the wider nanodisc sample (Fig. 2c). Second, local resolutions vary across our higher-resolution cryo-EM map of Doa10. The density for the scaffold is around 3 Å resolution, while the gate helices are resolved at much lower resolution (Supplementary Fig. 3a). This presumably reflects heterogeneous positioning of the gate compared to the rigid scaffold domain. Third, superposition of Doa10 coordinates derived from cryo-EM or AF prediction reveals a 7–9 Å displacement of the gate helices away from the scaffold domain in the predicted model (Fig. 5b).

Several MARCH6 mutants, exhibiting both defective and GOF phenotypes, are predicted to line the membrane channel interior (Fig. 3b, magenta). We hypothesized that the membrane channel could be a site for SQLE engagement, based on mutational effects, and its varying width, hydrophobicity, and location relative to the RING domain (Supplementary Fig. 10a, b). We modeled both the complex with full-length SQLE, or with its N-terminal 100 residues (SQLE-N100). SQLE-N100 was previously shown to be sufficient for recognition and ubiquitin-dependent degradation by MARCH6[30,31]. SQLE's N-terminal helices were placed inside MARCH6's membrane channel, although the models differed in orientation of the two SQLE helices (Fig. 5c). Defective MARCH6 mutants predicted to be near the SQLE N-terminus include MARCH6^Y92V, MARCH6^Y96V, and MARCH6^L140A/L747A. The strongest of our GOF mutants, MARCH6^Q866A/Q869A, is also predicted to be close to the bound substrate.

The model suggests that mutations lining the channel could affect direct binding and/or orientation of the substrate relative to the ubiquitylation active site. We thus surveyed these mutants by performing FLAG-immunoprecipitation for cell lines expressing FLAG-tagged MARCH6 variants followed by Western blotting against endogenously bound SQLE (Fig. 5d). This revealed a range of effects. The decreased amounts of endogenous SQLE co-precipitating with MARCH6^Y96V and MARCH6^Y92V/Y96V compared to MARCH6^WT, suggest that defective SQLE degradation in the Y96V mutant background can in part be explained by weakened interaction between SQLE and this

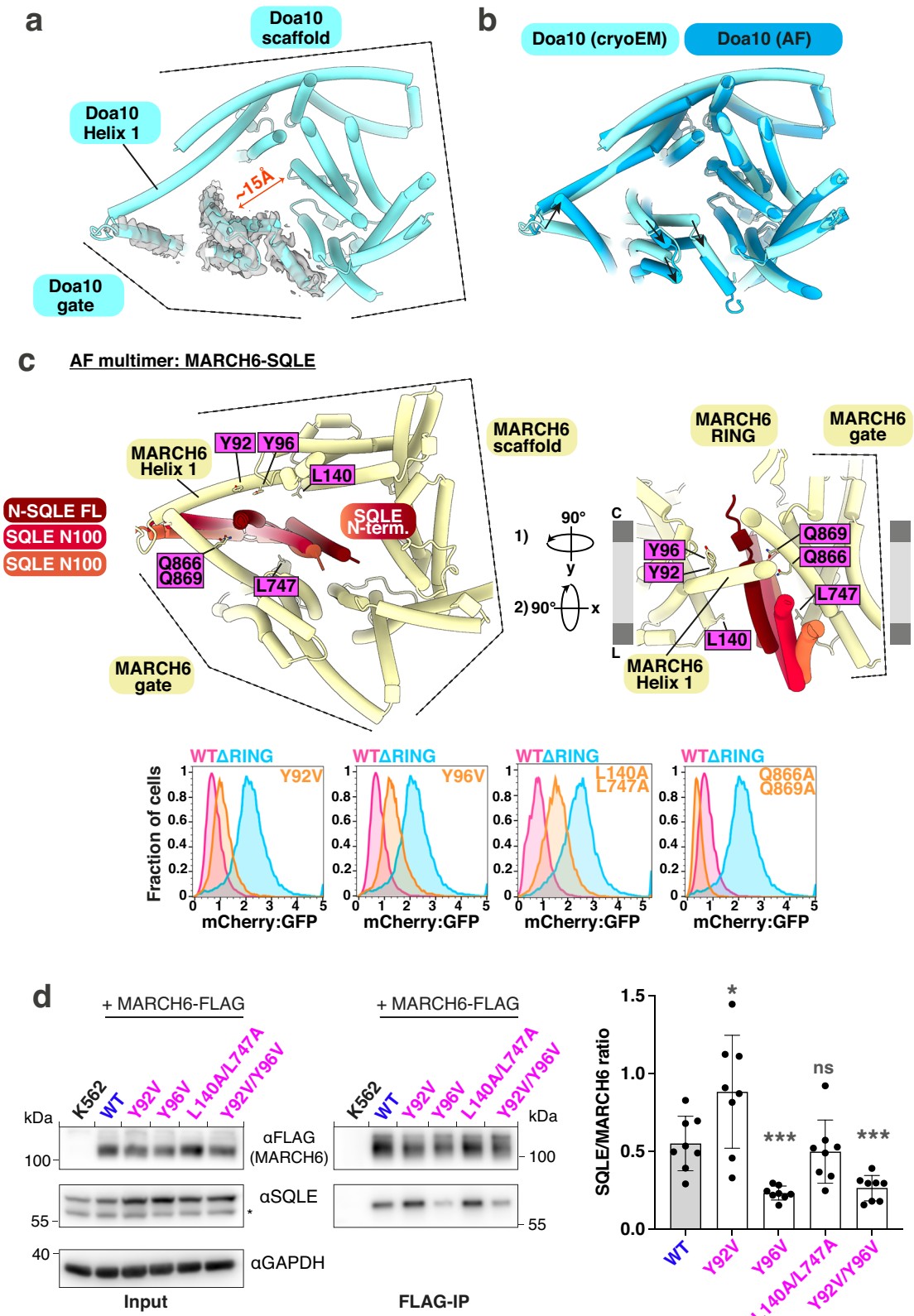

region of the MARCH6 membrane channel. Meanwhile, the other mutations maintained interaction and thus may affect capacity for SQLE ubiquitylation and turnover.

Our discovery of both SQLE stabilizing and destabilizing channel mutations in MARCH6 raises the possibility that the properties of this potential substrate engagement site have been adjusted for different substrates. Indeed, the MARCH6[Y96V] mutation, which shows weakened

interaction with SQLE, has contrasting effects on the levels of endogenous SQLE (a direct ubiquitylation substrate) and PLIN2 (whose levels are indirectly affected by MARCH6[39]). Both immunoblotting (Supplementary Fig. 7) and total proteomics (Supplementary Fig. 11) reveal SQLE stabilization and PLIN2 destabilization in the MARCH6[Y96V] cell line compared to MARCH6[WT]. We also note that the channel mutants displaying defective SQLE degradation express at levels

**Fig. 5 | A flexibly occluded membrane channel is the predicted site of SQLE engagement. a** High-resolution Doa10 cryo-EM structure with overlaid gate helices density. A central channel diameter of approximately 15 Å is indicated, measured between the center of gate Helix 13 and the start of scaffold Helix 7. **b** Overlay of Doa10 cryo-EM model (light blue) and Doa10 AF prediction (dark blue). Black arrows show gate helices movement away from the scaffold in the AF model. **c** Modeling a predicted MARCH6-SQLE complex. *Top*: AF multimer predicted complex between MARCH6 and SQLE (FL or SQLE-N100), with three models aligned on MARCH6, showing only SQLE-N100 for clarity (dark red: SQLE FL, red/orange: two predictions for positioning of the SQLE-N100 isolated N-terminus). Only the MARCH6 model from a single AF prediction is displayed for clarity, as E3 aligns almost identically across models. Membrane channel-lining residues are marked with magenta labels. *Bottom*: Flow cytometry panels comparing SQLE reporter levels in MARCH6-depleted cells re-expressing MARCH6$^{WT}$, MARCH6$^{\Delta RING}$,

or four MARCH6 variants with mutations outlined above (*top*). Relative mCherry fluorescence normalized to GFP as an expression control presented as a histogram. Representative result shown from six independent biological replicates. **d** Mutants within the MARCH6 membrane channel decrease SQLE binding. FLAG-immunoprecipitation of K562 cells expressing MARCH6-FLAG WT or four channel-lining mutants. *Left*: Representative Western blot against MARCH6-FLAG, SQLE and GAPDH (only input) for the whole cell lysate (input) and FLAG-immunoprecipitation (FLAG-IP). *Right*: Densitometric quantification of the SQLE/MARCH6 ratio for the FLAG-pulldown for eight independent biological replicates. Pairwise comparison of the mutants with the WT sample were done using a paired *t* test and significance levels are shown in grey above each mutant. *p* values: 0.0245 (Y92V), 0.0008 (Y96V), 0.5339 (L140A/L747A), 0.0009 (Y92V/Y96V). Source data provided as Source Data file.

similar to MARCH6$^{WT}$, indicating that these are likely not misfolded (Supplementary Fig. 6a). By contrast, GOF mutant MARCH6$^{Q866A/Q869A}$ accumulates compared to MARCH6$^{WT}$, which may partially explain the increased degradation of SQLE.

## Lipid-binding sites within the scaffold domain are structurally poised to influence E3 ligase function

The cryo-EM structure of Doa10 showed a key feature not available from AF predictions: a belt of lipid densities encircling the scaffold domain (Fig. 2e). Amongst these, four phospholipid molecules were firmly intercalated in the scaffold (Figs. 2f, 6a, Supplementary Fig. 3c). Residues interacting with lipids at three of these positions are structurally conserved from yeast to human E3 homologs. At the first position (Lipid 1), Doa10 K154 in Helix 1 contacts the lipid phosphate head group (Fig. 6b). The corresponding MARCH6 K116 is predicted to be structurally preserved at the same Helix 1 position, participating in a network of interactions including R113. The MARCH6$^{K116E}$ mutation results in a defect in SQLE reporter degradation without reducing mutant expression levels (Supplementary Fig. 6a). Mutation of the neighboring R113 (MARCH6$^{R113E}$), predicted to further "anchor" Helix 1 through interaction with D142 and S135, causes an even more significant SQLE degradation defect while also slightly increasing mutant levels. A second lipid binding site (Lipid 2) in Doa10 features F708 packing against the acyl chains of the bound phospholipid (Fig. 6c). This phenylalanine is conserved in MARCH6, and removal of the bulky hydrophobic side chain in the mutant MARCH6$^{F453A}$ leads to a defect in SQLE degradation without affecting mutant levels. In addition to these two positionally conserved lipid binding sites just described, we mutated a third potential MARCH6 scaffold lipid binding site (Lipid 3), corresponding to Doa10 K625 which interacts with a lipid phosphate group (Fig. 6d). Unlike the Lipid 1 and Lipid 2 sites, neither MARCH6$^{K374A}$ nor MARCH6$^{K374E}$ showed significant SQLE degradation defects.

While future structural studies will be necessary to confirm whether these sites in MARCH6 bind lipids, the observation of at least two defective MARCH6 mutants at these positions suggests functional importance of these residues. Notably, Helix 1 is positioned to mediate long-range conformational coupling of the lipid-binding scaffold to the gate helices and the catalytic RING domain.

## Discussion

Our integrated structural approach uncovered a conserved topology of Doa10/MARCH6 E3 ligases. Taken together with comprehensive mutagenesis, our data point to three structural features with key functional roles: 1) the RING domain binding the Ubc6/UBE2J2-ubiquitin conjugate and RING-adjacent regions positioning this catalytic assembly over the central channel; 2) the channel, which exhibits flexibility and appears to mediate membrane substrate engagement; and 3) lipid binding by the scaffold.

Positioning of the RING domain above the central channel seems crucial for ubiquitylation, as seen in the cryo-EM model of Doa10, the AF predicted ternary E3-E2-ubiquitin complex and mutational probing of this ternary model. Unlike many E3 ligases[9,58], the Doa10/MARCH6 RING domain is not flexibly tethered to the remainder of the structure. Instead, it is positioned by a conserved β-sheet formed by the residues downstream of the RING domain and the C-terminus. Our structure-wide screening across MARCH6, as well as previous targeted mutagenesis of Doa10[3,43], showed the importance of this pillar-like structure. Moreover, one side of the β-sheet connects via Helix 1 to the lipid-bound scaffold, and the other side connects to Helix 15 in the flexible gate. These functionally-important regions are predicted to be involved in substrate engagement. Thus, the overall Doa10/MARCH6 architecture interconnects the key functional elements (Fig. 7a).

In addition, the conformational flexibility of parts of the Doa10/MARCH6 membrane domain, as observed by cryo-EM and AF predictions, could enable Doa10/MARCH6 regulation of diverse substrates. It seems that efficiency of substrate engagement for ubiquitylation could be governed by: 1) the modulation of membrane channel dimensions, and potentially substrate access, via the flexibly-tethered gate, and 2) specific properties of the membrane channel lining. Also, given the diversity of ERAD substrates, it seems that there could be more than one substrate-binding modality, for example if the gate helices themselves could fold into the central channel to create a neo substrate-binding surface. Meanwhile, it seems that the intricate connections to the RING domain would allow for subtle reorientation of the ubiquitylation active site in response to substrate and lipid binding, for both modification of diverse substrates and regulation in response to lipid interactions.

Notably, MARCH6 regulates cholesterol homeostasis, a process which is on many levels tightly controlled by cholesterol or other lipids[24,59–66]. Indeed, cholesterol potentiates degradation of the SQLE reporter in our flow-cytometry screen. Compellingly, we also found well-resolved phospholipids interacting with the scaffold in our Doa10 structure. Our finding that mutation of these sites stabilizes SQLE, without diminishing MARCH6 protein levels, raises the exciting possibility that substrate regulation might be tied to sensing of metabolic signals within the lipid bilayer via MARCH6 conformation.

Our cryo-EM data for Doa10 now provides the third structure of a transmembrane E3 ubiquitin ligase. Structural comparison sheds light on their similarities and differences (Fig. 7b). The landmark structure of the yeast Hrd1 complex[9], which mediates degradation and retro-translocation of luminal and membrane ERAD substrates, showed Hrd1 and its partner protein Der1 distorting the membrane by deformation of the lipid bilayer and by thinning of the membrane. Interestingly, Doa10 reconstituted in lipid nanodiscs showed a similar, albeit less pronounced, effect (Supplementary Fig. 12). The second structure, also from the Rapoport lab, revealed the Pex transmembrane E3 ligase complex[67]. This complex forms an aqueous pore important for substrate retrotranslocation through the peroxisomal membrane in a

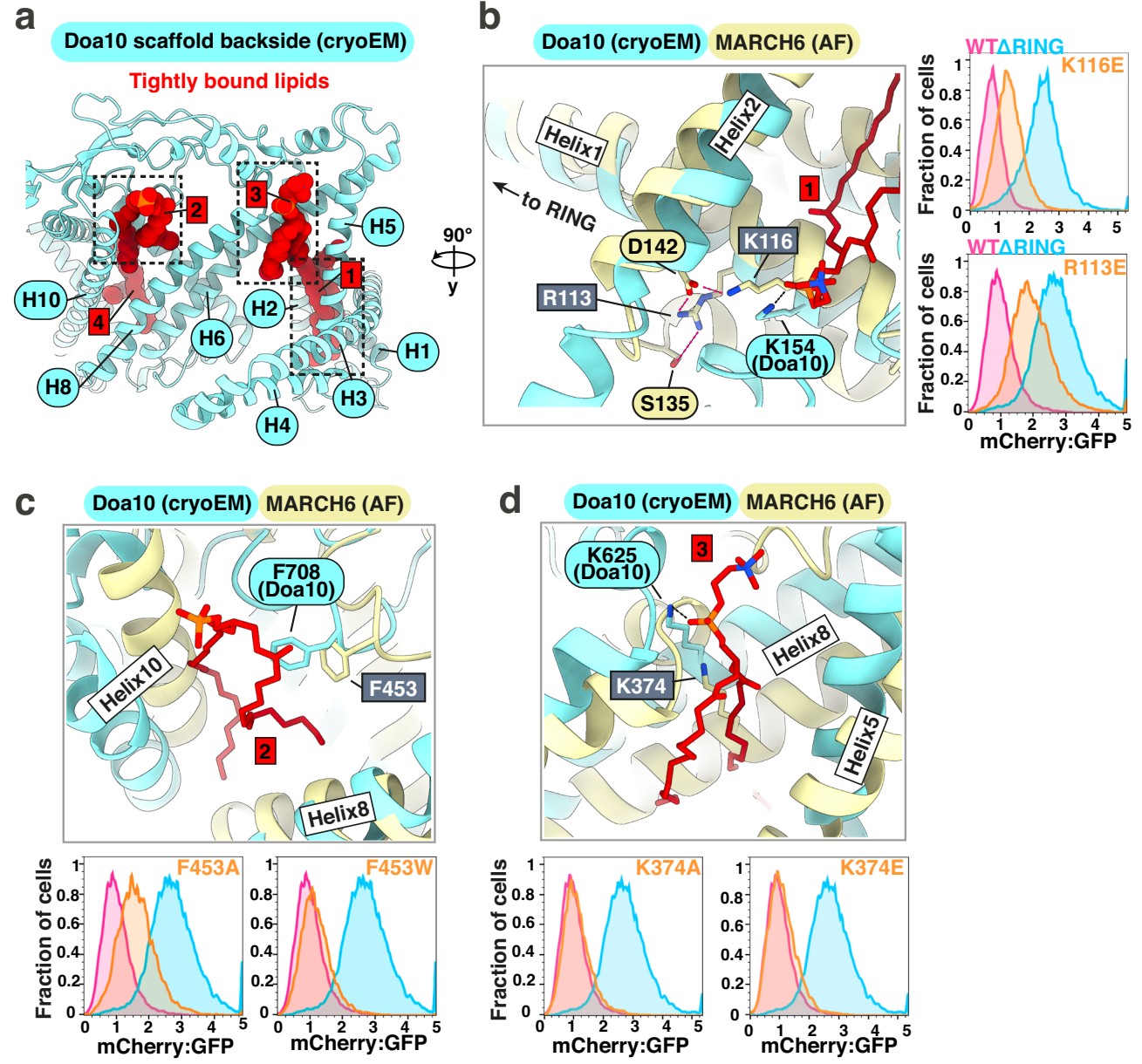

**Fig. 6 | Lipid binding sites in Doa10 are potentially conserved in MARCH6 and may partake in regulatory functions. a** Backside view of the scaffold domain in the Doa10 cryo-EM model. Well-resolved phospholipids shown as dark red spheres. Scaffold helices involved in lipid binding are indicated in light blue. Lipids are numbered, with dashed boxes around three lipid binding sites discussed in the text. **b** Scaffold Helix 1 and Helix 2 form a lipid headgroup binding site in Doa10. *Left*: Overlay of Doa10 cryo-EM structure and AF predicted model for MARCH6 at lipid binding site (Lipid 1). Doa10[K154] in Helix 1 interacts with the phosphate headgroup (black dashed line), with MARCH6[K116] in a structurally conserved position. AF prediction shows salt bridge network (magenta dashed lines) involving D142 in Helix 2, R113 in Helix 1, and S135 between Helix 1 and Helix 2. *Right*: Flow cytometry panels comparing SQLE reporter levels in MARCH6-depleted cells re-expressing MARCH6[WT], MARCH6[ΔRING], MARCH6[K116E], or MARCH6[R113E]. Histogram depiction of normalized mCherry:GFP fluorescence ratio. Representative shown from six independent biological replicates. **c** Doa10 Helix 10 and Helix 8 accommodate a bound lipid on the cytoplasmic membrane leaflet. *Top*: Doa10[F708] sidechain makes hydrophobic contact with Lipid 2 acyl chains. Predicted MARCH6 model shows conserved loop, with MARCH6[F453] sidechain protruding into a cavity where the lipid is resolved in Doa10. *Bottom*: Flow cytometry panels comparing SQLE reporter levels in MARCH6-depleted cells re-expressing MARCH6[WT], MARCH6[ΔRING], MARCH6[F453A], or MARCH6[F453W]. Histogram depiction of the normalized mCherry:GFP ratio. Representative result shown from six independent biological replicates. **d** *Top*: Depiction of Lipid 3 forming a hydrogen bond with Doa10 K625 in Helix 8. The conserved K374 of MARCH6 is shown in light yellow. *Bottom*: Flow cytometry panels comparing SQLE reporter levels in MARCH6-depleted cells re-expressing MARCH6[WT], MARCH6[ΔRING], MARCH6[K374A], or MARCH6[K374E]. mCherry fluorescence normalized to GFP as an internal expression control in histogram depiction. Representative result shown from six independent biological replicates.

manner similar to the Sec61 translocon in the ER[56,68,69]. Notably, our structural data also reveal a central channel in Doa10 and MARCH6. However, the Doa10/MARCH6 channel is hydrophobic and wider. It seems that the nature of the channel correlates with type of substrate, which for Doa10/MARCH6 includes membrane and cytosolic proteins but not luminal clients. The combination of membrane distortion, a

lipophilic channel, and conformational heterogenous regions with the potential to rearrange, suggests that each property may establish a variety of binding sites for different substrates.

Our study relied on combining structural and functional data across organisms. In synthesizing our structural analysis on yeast Doa10 with the results of a mutagenesis campaign for human

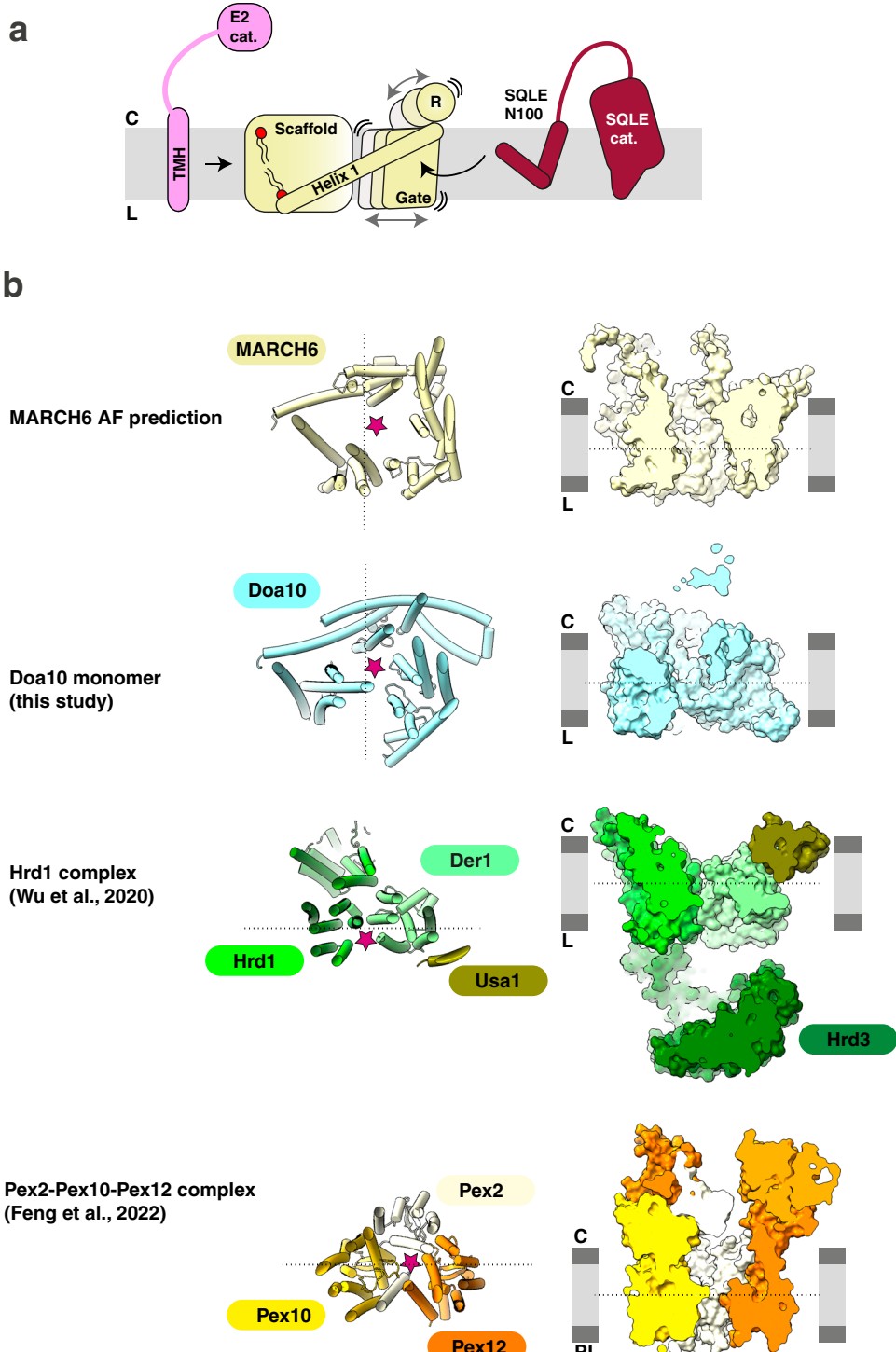

**Fig. 7 | Model for SQLE engagement and ubiquitylation by MARCH6. a** MARCH6 consists of a rigid scaffold domain linked to a flexible gate domain by the nearly perpendicular transmembrane Helix 1. The catalytic RING domain is supported above the membrane by N- and C-terminal elements, including residues adjacent to Helix 1 and the final gate Helix 15, forming a pillar-like beta-sheet. UBE2J2 TMH binding site is proposed on the backside of the MARCH6 scaffold, with SQLE-N100 engagement potentially occurring inside the E3 membrane channel near Helix 1 and 15. An arch-shaped lateral opening is formed by Helix 1 and 15, allowing SQLE-N100 entry into the membrane channel and potentially regulating the opening's width by the gate domain's swiveling. SQLE-N100 is positioned within the MARCH6 membrane channel for ubiquitylation, with sites aligned towards the cytosolic ubiquitylation active site. Binding of UBE2J2 TMH and catalytic domains to MARCH6 may create a temporarily rigid E3-E2-Ub-substrate complex, facilitating ubiquitin transfer to SM-N100 acceptor sites. Scaffold Helix 1 anchoring by scaffold elements or bound lipids may influence efficient conformational coupling between scaffold, gate, and RING domains of the E3. **b** Structural comparison of Doa10/MARCH6 with Hrd1 complex[9] (PDB: 6VJZ) and Pex complex[67] (PDB: 7T92). The left side depicts the structures of the E3 ligases in cartoon representation from the cytosol (top view) with a star highlighting the membrane channel. On the right side, a slice through the E3 ligases in surface representation from the membrane (side view) is shown. While MARCH6 and Doa10 form a hydrophobic membrane channel, the Pex complex forms a narrower aqueous pore and the Hrd complex assembles in a 'half-circle' leading to distortion and thinning of the membrane.

MARCH6, bolstered with in vitro assays and co-immunoprecipitation, the data extend our understanding to the human E3 ligase. We envisage that this strategy could enable structure-function studies of other transmembrane E3s[36,63,70]. Although E3 ligases are often considered to have distinct substrate binding and catalytic domains, our data for Doa10/MARCH6 suggest that interconnections between these functionalities and lipid binding sites could orchestrate metabolic signals, substrate binding, and ubiquitylation activity for transmembrane E3 ligases.

## Methods

### Cloning and plasmid construction for vectors used for recombinant expression

All constructs are listed in the Supplementary Data 1. All constructs were prepared using standard molecular biology techniques and Gibson assembly[71]. Doa10 and Ubc6 WT and mutants were cloned into pLib vector. Since the DNA of yeast Doa10 is toxic for *E. coli*[16,19,44], the Doa10 gene was codon optimized for expression in *Trichoplusia ni* High-Five insect cells and ordered by IDT (integrated DNA technology) with Gibson overhangs. Full-length Doa10 and mutants contained a C-terminal monomeric enhanced green fluorescent protein (mEGFP) followed by an ALFA-tag[72]. The mEGFP-ALFA-tag was cleavable by a HRV3C protease cleavage site between the Doa10-C-terminus and the mEGFP. The Doa10 construct used for sybody selection carried an additional N-terminal Avi-tag for selective biotinylation. Full-length Ubc6 was lifted from yeast genomic DNA and inserted into pLIB vector bearing a C-terminal TwinStrep-tag. The catalytic Cysteine (C87A) was mutated to Alanine to generate the catalytically dead E2, which was used for all studies (except in vitro *assays*) in this paper due to higher expression levels. Bacmids were generated in EMBacY *E. coli* cells for subsequent transfection of Sf9 insect cells (ThermoFisher Scientific).

For in vitro discharge assays, the catalytic (cat.) domain of UBE2J2 (2-170) and the RING domain of MARCH6 (2-71) were cloned into pGEX vectors. The soluble domains were tagged with an N-terminal GST-tag followed by a HRV3C cleavage site. Mutations of these domains were introduced with Gibson quickchange and oligos are listed in the Supplementary Data 2. Ubiquitin WT and I44A were cloned into pET3b without any tags.

### Protein expression and purification of soluble domains

Soluble domains of MARCH6 and UBE2J2, and ubiquitin for in vitro discharge assays were expressed in *E. coli* Rosetta. After induction with 0.3 mM IPTG, the proteins were expressed at 18 °C overnight. The cell pellets for the RING and E2 cat. domain were resuspended in buffer S (30 mM Tris pH 8.0, 300 mM NaCl, 5 mM DTT) supplemented with protease inhibitor cocktail (EDTA-free) and 0.001 mg/mL benzonase and lysed by sonication on ice. After centrifugation, the soluble lysate fraction was incubated with glutathione-agarose beads and washed with buffer S. Incubation with 0.05 mg/mL HRV3C protease at 4 °C overnight cleaved the N-terminal GST tag and eluted the proteins from the beads. The RING domain variants were then concentrated and loaded onto a Superdex 75 10/300 Increase size exclusion column (GE Healthcare) with buffer D (25 mM HEPES pH 7.5, 100 mM NaCl, 1 mM TCEP). The UBE2J2 cat. domain constructs were diluted in buffer S (pH adjusted to 6.8) and subsequently loaded onto a HiTrap SP HP cation exchange column (Cytiva). A gradient of 50–500 mM NaCl eluted the protein from the column and E2 containing fractions were concentrated and polished in a final size exclusion chromatography step using a Superdex 75 10/300 Increase column (GE Healthcare) in buffer D.

Cells expressing ubiquitin (WT and I44A) were resuspended in buffer U (50 mM Tris pH 7.5, 200 mM NaCl, 5 mM DTT) supplemented with 2.5 mM PMSF. After cell lysis using an emulsiflex high pressure homogenizer (Avestin), the lysate was cleared by centrifugation. The

drop-wise addition of glacial acetic acid to the soluble fraction to adjust the pH to 4.5-5 precipitated a major fraction of the proteins in solution except for ubiquitin withstanding low pH values. Further centrifugation (15,000 x g, 20 min) separated the precipitated protein fraction. Next, dialysis into 25 mM sodium acetate pH 4.5 with 5 mM DTT was performed overnight, followed by another centrifugation step to separate remaining precipitation. The soluble fraction was incubated with Sepharose S resin (Cytiva) and washed with 25 mM sodium acetate, 5 mM DTT and 75 mM NaCl. Elution of the protein was achieved by the addition of 250 mM NaCl to the sodium acetate buffer. Concentrated protein was loaded onto a Superdex 75 HiLoad 60/1600 pg (GE Healthcare) in buffer D.

### Protein expression and purification of membrane proteins

Baculovirus of Doa10 and Ubc6 constructs were amplified in Sf9 insect cells and then used to transfect *Trichoplusia ni* Hi5 cells (ThermoFisher Scientific) for expression of four different complexes: Avi-Doa10-mEGFP-ALFA (for ELISA validation during sybody selection), Doa10-mEGFP-ALFA (for structural studies), Avi-Doa10-mEGFP-ALFA and Ubc6(C87A)-Strep (for sybody selection), Doa10-mEGFP-ALFA and Ubc6(C87A)-Strep (for competition step in sybody selection and structural studies). Cell lysis, solubilization and incubation with ALFA nanobody-beads were done as followed for all complexes. Cells were harvested after 4 days of expression and resuspended in hypotonic buffer (50 mM Tris pH 7.5, 5 mM NaCl, 1 mM MgCl₂) supplemented with 3 mM DTT, EDTA-free protease inhibitor cocktail and 0.001 mg/mL benzonase to lyse the cells. After incubation on ice for 20 min, the membrane fraction was pelleted by centrifugation at 106,000 x g for 1 h. The insoluble fraction was resuspended in buffer A (30 mM Tris pH 7.5, 300 mM NaCl, 3 mM DTT) supplemented with EDTA-free protease cocktail and 0.75% (w/v) glycol-diosgenin (GDN, Anatrace) and incubated at 4 °C for 3 h. The insolubilized material was separated by another centrifugation step at 106,000 x g for 1 h. The supernatant was incubated with ALFA nanobody coupled Sepharose beads (CNBr-activated Sepharose 4B (Cytiva), for coupling of the beads the manufactures instructions were followed) at 4 °C for 1 h. The beads were thoroughly washed with buffer A supplemented with 0.02% GDN to wash away unbound protein. Additionally, the beads were washed with 30 mM Tris pH 7.5, 250 mM NaCl, 50 mM KCl, 3 mM DTT, 0.02% GDN and 3 mM ATP to release any bound chaperones.

The bound Avi-Doa10 sample was subsequently incubated with buffer B (buffer A supplemented with 0.01% GDN, 25 mM MgCl₂, 25 mM ATP, 1 μM BirA and 3 mM biotin) supplemented with 0.01 mg/mL HRV3C protease at 4 °C overnight to elute the protein from the beads and to biotinylate the Avi-tag. The biotinylated protein was concentrated using a 100 kDa cut-off Amicon centrifugal filter (used for all concentration steps) and loaded onto a Superose 6 Increase 10/300 GL column (GE Healthcare) with buffer C (25 mM HEPES pH 7.5, 100 mM NaCl, 3 mM DTT) supplemented with 0.01% GDN. Peak fractions containing Avi-Doa10 were pooled, concentrated and used for ELISA validation of the sybodies.

For structural studies, the bound Doa10 protein was eluted with buffer A supplemented with 0.02% GDN and 0.05 mg/ml HRV3C protease at 4 °C for 2 h. The eluted protein was concentrated and reconstituted in MSP2N2 (available from the MPIB core facility) (see below).

For sybody selection, nanodisc reconstitution and structural studies, the bound Doa10-Ubc6 or Avi-Doa10-Ubc6 complex were eluted from the ALFA nanobody coupled Sepharose beads by incubating the beads with buffer A supplemented with 0.02% GDN and 0.05 mg/ml HRV3C protease at 4 °C for 2 h. Then the sample was incubated with Strep-Tactin resin at 4 °C for 30 min. The beads were washed with buffer A supplemented with 0.02% GDN to wash away free Doa10. The protein complex was eluted with buffer A supplemented with 0.02% GDN and 2.5 mM desthiobiotin and afterwards concentrated using a 100 kDa cut-off Amicon centrifugal filter.

For biotinylation of the Avi-tag, the Avi-tagged Doa10-Ubc6 complex was incubated with buffer B at 4 °C overnight. Afterwards, the biotinylated protein was loaded onto a Superose 6 Increase 10/300 GL column (GE Healthcare) in buffer C supplemented with 0.01% GDN.

The Doa10-Ubc6 complex without the Avi-tag was directly loaded onto the SEC column skipping the biotinylation step. Fractions containing Doa10 and Ubc6 were pooled, concentrated and used for sybody selection.

## Reconstitution into nanodisc

For nanodisc reconstitution, concentrated Doa10-Ubc6 and biotinylated Avi-Doa10-Ubc6 were incubated with MSP1E3D1 (available from the MPIB core facility) and yeast polar lipid extract (Avanti Polar Lipids) prior to size exclusion chromatography (SEC). The Doa10-Ubc6 complexes, MSP1E3D1 and lipids were mixed in a ratio of 1:4:50 and incubated on ice for 1 h. For structural studies, sybody 37 (sb37) was added to the mixture in 4-fold access. To achieve nanodisc reconstitution, the mixture was incubated with Bio-Beads SM2 (Bio-Rad). After 2 h, Bio-Beads were removed and fresh Bio-Beads were added for another 2 h. To remove free nanodiscs from reconstituted proteins, a final SEC run was done using Superose 6 Increase 10/300 GL column in buffer C. Peak fractions of the biotinylated Doa10-Ubc6 complex were concentrated to 5 μM and used for ELISA validation of the sybodies. Peak fractions of the Doa10-Ubc6-sb37 complex reconstituted in MSP1E3D1 were concentrated to 4.2 mg/mL and directly used for grid preparation.

In addition to MSP1E3D1 reconstitution, Doa10 was also reconstituted in MSP2N2 (available from the MPIB core facility) using a protein to MSP2N2 to lipid ratio of 1:3:50. The reconstitution into MSP2N2 followed the same protocol as MSP1E3D1 reconstitution (see above) but the protein was concentrated to 5.2 mg/ml.

## Selection of E3 ligase specific sybodies

A useful method for aiding transmembrane protein structure determination is inclusion of a bound affinity reagent[50,73]. The biotinylated Avi-Doa10-Ubc6 complex in GDN was used for sybody selection. Biotinylation efficiency was tested by capturing the biotinylated protein with streptavidin-coated magnetic beads (Promega). The sybody selection was executed following the protocol published by the Seeger lab[51]. Briefly, three selection rounds were done using the synthetic nanobody (sybody) library generated by the Seeger lab[50]. First, 50 nM of biotinylated Avi-Doa10-Ubc6 complex in GDN were used for ribozyme display. Then two rounds of phage display were performed with 50 nM of biotinylated Avi-Doa10-Ubc6 complex in GDN. To increase the number of high-affinity binders, a competition step with 50 μM of non-biotinylated Doa10-Ubc6 complex in GDN was added to the second phage display. 95 sybodies were expressed for ELISA screening and 48 ELISA hits were analyzed by Sanger sequencing. Unique sybody hits were used in a subsequent ELISA using biotinylated Avi-Doa10 and Avi-Doa10-Ubc6 in GDN as well as Avi-Doa10-Ubc6 in MSP1E3D1 to identify sybodies specific for Doa10, which would not be sterically hindered by Ubc6 or MSP1E3D1 and its associated lipids. 12 unique sybodies binding the Doa10 complex reconstituted in lipid nanodisc were then expressed in *E. coli* and purified using periplasmic expression, Ni-affinity chromatography and SEC. Binding to Doa10 was validated with SEC experiments. Sybody 37 (sb37) was used for structural studies of the Doa10-Ubc6 complex.

## Single-particle cryo-EM sample preparation and data acquisition

Quantifoil holey-carbon cryo-EM grids (R1.2/1.3, 200 Cu mesh) were glow-discharged and 4 μL of freshly purified and concentrated Doa10-Ubc6-sb37 in MSP1E3D1 was applied to the grids. Sample application was done using a Vitrobot Mark IV (Thermo Fisher Scientific) with a wait time of 6 s, a blot force of 3, a blot time of 4 s, at 100% humidity and 4 °C. Grids were plunge-frozen into liquid ethane. High-resolution

cryo-EM data were collected on a Titan Krios Cryo-Transmission Electron Microscope (Cryo-TEM) operating at 300 kV equipped with a Gatan BioQuantum K3 direct electron detector in counting mode and a Quantum-LS energy filter. Videos were collected at a nominal magnification of x 105,000 corresponding to a pixel size of 0.8512 Å/px. A defocus range between −0.7 and −2.8 μm was chosen and 40 frames were collected with a total electron dose of ~70 e⁻/Å². Every data collection was done using SerialEM v3.8.0[74]. Low-resolution data of the Doa10-Ubc6 complex in MSP1E3D1 and Doa10 in MSP2N2 were collected on a Glacios Cryo-TEM operating at 200 kV equipped with a Gatan K2 Summit direct electron detector (counting mode). A magnification of x 22,000 was used corresponding to a pixel size of 1.885 Å/px. 40 frames were collected for 0.4 s with a total exposure of ~70 e⁻/Å² and a defocus range between −1.2 μm and −3.3 μm.

## Data processing

The high-resolution dataset compromised of 14.635 videos which were imported into Relion 3.1.1[75] and motion corrected using Relion's own implementation. Contrast transfer function (CTF) estimation was done using CTFFIND-4.1[76]. Micrographs were then imported into cryoSPARC v3.1[77] and manually curated to exclude micrographs collected on carbon. 2,406,662 particles were picked on 13,068 micrographs with cryoSPARC's blob picker. The particles were extracted with a box size of 304 px and Fourier cropped to 152 px. Next, particles were sorted with several rounds of 2D classification using default parameters in cryoSPARC except for 'number of online-EM iterations' set to 40 and 'batchsize per class' set to 500. After further classification using the cryoSPARC ab initio job, an initial model was generated with 106,325 particles. This initial model was used as a template for template picking of 4,805,403 particles. Particles were sorted with 2D classification, ab initio and heterogeneous refinement using the initial model and a model of an empty nanodisc as references. Default parameters were used except for 'refinement box size' = 152 Å, 'batch size per class' = 5000, 'initial resolution' = 9 Å. After several classification rounds, 159,788 particles were re-extracted with a box size of 304 px. One more round of heterogenous refinement was conducted before 123,143 particles were used for a locale CTF refinement job and several, iterative non-uniform refinement jobs[78] (initial low-pass resolution = 9 Å, non-uniform AWF = 2, batchsize epsilon = 0.01, dynamic mask far = 20 Å, optimize per-particle defocus, optimize per-group CTF parameters) resulting in an overall resolution of 3.58 Å. 3D classification or 3D variability jobs did not yield alternative conformations or complexes for the Doa10 cryo-EM sample.

The low-resolution dataset of the Doa10-Ubc6 complex in MSP1E3D1 consisted of 4,926 movies. The movies were processed similar to the high-resolution movies collected on the Titan Krios. A total of 2,350,910 particles were picked using Gautomatch v0.56 (K. Zhang, MRC Laboratory of Molecular Biology) and extracted with a box size of 160 px using Relion's extraction tool. The extracted particles were imported into cryoSPARC and submitted to several 2D classification runs. After the fifth classification, 541,982 particles were selected for classification with ab-initio job in cryoSPARC. A final batch of 64,043 particles was used for non-uniform refinement using the initial model of the ab-initio job. Two runs yielded the final model with a resolution of 6.73 Å.

For the low-resolution dataset of Doa10 in MSP2N2 2,196 movies were collected. A similar pipeline as for the high-resolution dataset was used for motion correction and CFT refinement. Afterwards a Topaz[79] model was trained in cryoSPARC because of a very high particle density on the grid. First, an initial batch of almost 3,000 particles were picked manually and extracted with a box size of 192 px. Those particles were submitted to 2D classification and used to train a Topaz model (best model: default parameters + 'expected number of particles: 500', 'use pretrained initialization', 'number of epochs: 5'). Particles were picked using the trained model and 663,574 particles were

extracted with a box size of 160 px. After one round of 2D classification, 536,794 particles were sorted using ab-initio in cryoSPARC. A final batch of 188,744 particles was used for two non-uniform refinement runs yielding a map with an overall resolution of 5.77 Å.

## Model building and refinement

A structure of Doa10 and Ubc6 were predicted using AlphaFold2[45]. The predicted model was docked into the density map and manually adjusted. Poorly resolved regions or sidechains with insufficient density were deleted. Density corresponding to lipids were used to build DOPC or 1,2-dipalmitoyl-glycero-3-phosphate (if there was no density for the head group). All model building and adjustments were done in coot v0.9.6[80]. Several real-space refinement jobs were done in phenix v1.19.2[81] using the sharpened non-uniform refinement map from cryoSPARC with subsequent visual verification in coot[80]. Visual representation of (protein) structures were prepared using ChimeraX v1.4[82].

## In vitro ubiquitin discharge assays

The discharge assays were performed in a pulse-chase format. First, the UBE2J2 cat. domain (WT or mutants) was loaded with ubiquitin. The loading reaction was set up with 20 mM MES pH 6.0, 100 mM NaCl, 10 mM MgCl$_2$, 10 mM ATP, 20 µM E2 and 0.1 µM E1 (or 0.2 µM E1 if I44A ubiquitin was loaded onto the E2). The reaction was started by the addition of 60 µM WT or I44A ubiquitin and incubated at room temperature (RT) for 30 min. Afterwards, loading by the E1 was quenched by the addition of 4 U apyrase (NEB) and incubation at RT for 5 min. The loaded E2-Ub sample was diluted with HEPES to adjust the pH to 7.5. A fraction of the sample was added to 5x SDS loading dye as the 0 min time point. The remaining sample was added to 30 µM MARCH6 RING domain (WT or mutants) to start the discharge of ubiquitin from the UBE2J2 cat. domain. Six time points at 2, 5, 10, 15, 20, 30 min were taken and quenched with 5x SDS loading dye. Reactions were run using a 12% SDS-PAGE gel, stained with Coomassie and imaged using a Amersham 600 gel imager. All reactions were done in triplicates. Band intensities were quantified using ImageJ v2.0.0 and graphically summarized using GraphPad Prism v10.0.2. Source data underlying the quantification are provided as Source Data file.

## AlphaFold structure prediction

We generated several models using the open source code of AlphaFold2[45] or AlphaFold multimer[52]. WT sequences for MARCH6, Ube2J2, Doa10, Ubc6 and ubiquitin were downloaded from uniprot and used for the prediction. The parameter used for predictions were a maximum release date of 2022-10-01, using full_dbs and asking for 25 models. The predictions for the Doa10, the Doa10/Ubc6 complex, MARCH6, MARCH6/UBE2J2/ubiquitin and MARCH6-RING/UBE2J2-cat./ubiquitin did mostly align over all 25 models. For the MARCH6/SQLE or MARCH6/SQLE-N100 predictions <35% of the models predicted the N-terminus of SQLE outside of the gate cavity of MARCH6.

## Dual fluorescence MARCH6 client reporter cell lines

The dual fluorescent reporter for SQLE was introduced lentivirally into a K562 cell line containing UCOE-SFFV-Zim3-dCas9-P2A-hygro[47]. The cell line was a generous gift from Alina Guna. A volume of virus was used for spinfection (1000 x g, 2 h) that produced ~20–30% GFP fluorescent cells used for sorting. Cells were maintained in RPMI 1640 medium (Gibco) supplemented with 0.01 mg/mL hygromycin. The full-length sequence for SQLE (uniprot: E5RJH9) was cloned into a pKDP110 vector with a C-terminal mCherry-P2A-GFP tag[48]. 48 h post-transduction, GFP/mCherry-positive cells were sorted on a BC CytoFLEX SRT (Beckman Coulter, MPIB Facility Sorter). The plasmid sequence for the reporter construct is available in the Supplementary Data 2. Regular mycoplasma tests were done to ensure no mycoplasma contamination of the cells.

## CRISPRi knock-down of MARCH6

Knock-down of MARCH6 was achieved using CRISPRi technology as described previously[47,83]. The two top scoring guides for MARCH6[83] (5′-ggtacgctcaggtgcgagag-3′ and 5′- ggaggggcgggaacacctag-3′) were expressed from dual-guide lentiviral vectors bearing a PuroR-P2A-BFP selection marker. The plasmid sequence of this guide vector is available in the Supplementary Data 1, and dual-guide cloning was done as described previously[47]. A volume of virus was used for spinfection that produced ~20-30% BFP fluorescence, as titered using flow cytometry. 48 h post-transduction into K562 UCOE-SFFV-Zim3-dCas9-P2A-hygro cells (with or without fluorescent substrate reporters), selection was done using 1.5 µg/mL puromycin over 72 h. Fresh puromycin was added daily. Successful knock-down was validated via qPCR against MARCH6 mRNA and stabilization of endogenous SQLE by immunoblot[34,39].

## Re-introduction of wild-type of mutant MARCH6 into CRISPRi MARCH6 depleted cells

Lentivirus was used to re-introduce either wild-type or mutant MARCH6 into CRISPRi MARCH6 depleted K562 UCOE-SFFV-Zim3-dCas9-P2A-hygro cell lines with or without fluorescent reporter. Here, MARCH6 variants were expressed with a C-terminal FLAG-tag for immunoblot detection, driven by a SFFV promoter. The plasmid sequences for the WT and ΔRING rescue constructs are available in the Supplementary Data 2. All mutations were introduced using the MARCH6 WT construct with site directed mutagenesis. For selection and viral titering, the expression vector carries an IRES-miRFP680-P2A-BSR cassette (blasticidin resistance marker). To avoid multiple viral integrations per cells, a volume of virus was used for each mutant which produced 15-30% miRFP680 fluorescence. 48 h post-transduction into K562 UCOE-SFFV-Zim3-dCas9-P2A-hygro cells (with or without fluorescent substrate reporters), selection was carried out using 40 µg/mL blasticidin over 7 days. Fresh blasticidin was added to cultures on day one, day three and day five of selection.

## Flow cytometry analysis of MARCH6 fluorescent substrate reporters

Dual fluorescence MARCH6 substrate reporters were analyzed by flow cytometry using an Attune NxT flow cytometer. For each sample, ~15,000 live cells were analyzed. Following live cell gating based on FSC/SSC, cells were further gated on BFP (CRISPRi knock-down guide expressing) and then miRFP680 (ectopic MARCH6-FLAG variant expression). The gating strategy is visualized in Supplementary Fig. 13. Finally, the ratio of mCherry to GFP fluorescence was plotted for each mutant and control cell lines. Each flowcytometry panel presents the ratio for both controls (MARCH6 WT and ΔRING) and the respective mutant in histogram representation. Subsequent data analysis was performed using FlowJo software. Each MARCH6 mutant was screened in at least four biological replicates, with separate controls (wild-type or ΔRING) included for every replicate. For each biological replicate, we conducted fresh CRISPRi depletion, re-introduction of MARCH6 variants and their respective selection steps prior to the reporter stability screen. Flow cytometry data were analyzed using FlowJo v10.8.2 followed by statistical analysis in GraphPad Prism v10.0.2. Details for each mutant, including number of replicates, mean, standard error of mean (SEM) and P values of the comparison to the WT sample using one-way ANOVA are listed in the Supplementary Data 3. Source data used for analysis is provided as Source Data.

## Antibodies used for western blotting

The following antibodies were used for immunoblotting: anti-SQLE (Proteintech 12544-1-AP, rabbit, 1:500), anti-PLIN2 (Cell Signaling 45535, rabbit, 1:500), anti-FLAG (Sigma F1804, mouse, 1:2000), anti-GAPDH (Cell Signaling D16H11, rabbit, 1:2000), anti-mouse-HRP (Cell Signaling 7076, 1:5000), anti-rabbit-HRP (Cell Signaling 7074, 1:5000).

Each antibody was diluted using TBST (50 mM Tris, 150 mM NaCl, 0.1% Tween pH 7.6) containing 5% (w/v) milk.

## Western blotting of mammalian lysates

For each sample, 0.5 million cells were harvested and lysed in 120 μL cold RIPA buffer. After incubation for 30 min on ice, 20 μL of 5x SDS sample buffer was added and samples were heated at 37 °C for 30 min. Samples were stored at −80 °C until use. For immunoblotting, 10 μL of sample per well was separated on SERVAGel™ TG PRIME™ 12% (SERVA) and subsequently transferred onto PVDF membrane for 2 h at 250 mA using wet tanks at 4 °C. Membranes were blocked for 1 h at room temperature using 5% milk in TBST and incubated with primary antibody over night at 4 °C. The next day, membranes were washed with TBST and incubated with secondary HRP-conjugated antibody for 1 h at room temperature. Following washes with TBST, membranes were incubated with ECL solution and imaged for chemiluminescence using an Amersham 600 imager. Raw western blots are provided as Source Data.

## FLAG-pulldown of MARCH6 for SQLE binding assay

Lentivirus was used to introduce either wild-type or mutant MARCH6 into K562 UCOE-SFFV-Zim3-dCas9-P2A-hygro cell lines still expressing endogenous MARCH6. The introduced MARCH6 variants were expressed from the same expression vector as mentioned above under the regulation of a SFFV promoter with a C-terminal FLAG-tag for immunoblot detection. After selection of infected cells with blasticidin (as mentioned above) and monitoring of successful virus transfection by miRFP680 fluorescence, cells were expanded and a final number of 10 Mio cells were harvested by centrifugation (300 x g, 4 min). Cell pellets were resuspended in 900 μL buffer R (30 mM Tris pH 8.0, 200 mM NaCl, 2 mM DTT) supplemented with protease inhibitor cocktail and 0.001 mg/mL benzonase. The cells were lysed by incubation with 1% digitonin on ice for 30 min. Afterwards, the solubilized fraction was separated by centrifugation (21,000 x g, 15 min). For the Input western blot samples, 100 μL lysate was added to 25 μL 5x SDS-loading dye (reducing). 740 μL of the remaining cell lysate were incubated with 10 μL Pierce Anti-DYKDDDDK magnetic agarose (Thermo Scientific) slurry and incubated at 4 °C for 2 h. The beads were washed with buffer R with 0.1% digitonin. Finally, the beads were resuspended in 35 μL buffer R with 0.1% digitonin and 17.5 μL 3x Laemmli loading dye (non-reducing). Sample preparation and western blot procedure were done as described above. All samples were done in eight biological replicates. Band intensities were analyzed using ImageJ v2.0.0 and statistical analyses was conducted in GraphPad Prism v10.0.2. Source Data is provided as Source Data file.

## Quantitative PCR (qPCR)

For each sample, 0.5 million cells were lysed in 1 mL TRIzol (Thermo Fisher) and stored at −80 °C until use. To each sample, 200 μL Chloroform was added and mixed thoroughly by shaking, followed by centrifugation at 14,000 x g at 4 °C for 10 min. Of each sample, 400 μL of the colorless supernatant was added to 500 μL isopropanol in fresh tubes and the tubes inverted several times. Samples were incubated over night at −20 °C. The next day, tubes were centrifuged at 14,000 x g at 4 °C for 10 min. The supernatant was carefully removed and 500 μL 70% ethanol added, followed by centrifugation for 5 min. The ethanol was carefully removed and the pellets dried for 10 min at room temperature. Pellets were resuspended in 25 μL RNase free water at frozen at −20 °C for storage.

For reverse transcription, 1 μg of RNA for each sample was diluted to a total volume of 11 μL with RNase free water in fresh PCR tubes. Per tube, the following reaction components were added: 0.4 μL dT oligos (stock: 50 μM), 0.06 μL random hexamer primers (stock: 50 μM), 0.54 μL RNase free water. The reactions were first incubated for 4 min at 65 °C and then for 10 min on ice. To each tube, the following reaction components were added: 4 μL 5x Frist strand buffer (5x SSIV buffer), 2 μL DTT (stock: 0.1 M), 0.5 μL dNTPs (stock: 10 mM), 1.25 μL RNase free water, 0.25 μL Superscript reverse transcriptase (stock: 200 U/μL). Reactions were incubated for 2 h at 42 °C, after which 40 μL RNase free water were added and tubes vortexed. Primer pairs used for subsequent SYBR Green qPCR were as followed: GAPDH control (fw: 5'-GTTCGACAGTCAGCCGCATC-3', rv: 5'-GGAATTTGCCATGGGTGGA-3'), MARCH6 (fw: 5'-GACTGGTTACAAGTATTGGCACT-3', rv: 5'-CCGTTGAC AGCATATCTAATGGC-3'). qPCR reactions were set up as follows per tube: 0.2 μL primer fw, 0.2 μL primer rv, 5 μL BioRad Sso Mix, 2.6 μL RNase free water, 2 μL cDNA. For TaqMan qPCR, the following primers were used: GAPDH control (Hs02786624_g1, Thermo Fisher) and MARCHF6 (Hs01020084_m1, Thermo Fisher). TaqMan qPCR reactions were set up using per reaction 5 μL Taqman fast advanced MasterMix (Applied Biosystems), 0.5 μL primer, 0.5 μL RNase free water and 4 μL cDNA. Samples were run in duplicate on a BioRad CFX96 Real-Time System and analyzed using BioRad CFX Manager v2.1 software.

## Total proteome mass spectrometry analysis

We compared the total proteome of MARCH6^WT cells with cells expressing MARCH6^Y96V. For this K562 UCOE-SFFV-Zim3-dCas9-P2A-hygro cells with a MARCH6 knockdown were rescued with MARCH6^WT or MARCH6^Y96V as described above. The cells were grown at 37 °C for 48 h before a total of two million cells were harvested with centrifugation at 300 x g and washed with PBS. The cell pellet wasflash-frozen and stored at −80 °C. Three biological replicates were prepared for mass spectrometry. The frozen pellet was resuspended in 300 μL of SDC buffer (1% sodium deoxycholate, 40 nmM 2-chloroacetamide (Sigma-Aldrich), 10 mM tris(2-carboxyethyl) phosphine (TCEP; PierceTM, Thermo Fisher Scientific) in 100 mM Tris, pH 8.0) and incubated at 95 °C for 2 min followed by sonification for 10 min at 4 °C. Parts of the samples (50%) were incubated once more at 95 °C for 2 min followed by sonification for 10 min at 4 °C. These aliquots were diluted 1:2 with MS grade water (VWR) and supplemented with 1 μg of LysC and 3 μg of trypsin (Promega) for digestion at 37 °C overnight. The peptides were then acidified with trifluoroacetic acid and purified using SCX StageTips. Samples were vacuum dried and resuspended in 0.1% formic acid. Desalted peptides were loaded onto a 30-cm column (inner diameter: 75 μm; packed in-house with ReproSil-Pur C18-AQ 1.9-μm beads, Dr. Maisch GmbH) at 50 °C. Using the nanoelectrospray interface, eluting peptides were directly sprayed onto the benchtop Orbitrap mass spectrometer Q Exactive HF (Thermo Fisher Scientific).

The samples were separated by two buffers, buffer A (0.1% formic acid) and buffer B (0.1 formic acid, 80% acetonitrile), at 250 nL/min. A gradient of buffer B from 2%-30% over 120 min followed by 30–60% over 10 min and then to 95% over 5 min was used. Afterwards, buffer B concentration of 95% was maintained for another 5 min. A data-dependent mode with survey scans from 300 to 1750 m/z (resolution of 60000 at m/z = 200) was used, and up to 15 of the top precursors were selected for fragmentation at higher energy collisional dissociation (HCD with a normalized collision energy of value of 28). The MS2 spectra were recorded at a resolution of 15000 (at m/z = 200). AGC target for MS and MS2 scans were set to 3E6 and 1E5, respectively, within a maximum injection time of 100 and 60 ms for MS and MS2 scans, respectively, and the dynamic exclusion was set to 30 ms.

Raw data were processed using the MaxQuant computational platform[82] (version 2.2.0.0) with standard settings applied. Shortly, the peak list was searched against the reviewed human Uniprot database (downloaded in August 2021) with an allowed precursor mass deviation of 4.5 ppm and an allowed fragment mass deviation of 20 ppm. MaxQuant by default enables individual peptide mass tolerances, which was used in the search. Cysteine carbamidomethylation was set as static modification, and methionine oxidation and N-terminal acetylation as variable modifications. The Perseus software package version 2.0.9.0 was used for the data analysis[83]. Protein intensities were

log2-transformed and filtered to make sure that identified proteins showed expression in all biological triplicates of at least one condition and to avoid reverse hits and contaminants. The missing values were subsequently replaced by random numbers that were drawn from a normal distribution (width = 0.3 and down shift = 1.8). For volcano plots, we used permutation-based FDR, which was set to 0.05 in conjunction with an s0-parameter of 0.1 to determine the significance. The mass spectrometry proteomics data have been deposited to the ProteomeXchange Consortium[84] via the PRIDE[85] partner repository with the dataset identifier PXD047499.

## Statistical analyses

Data from at least four independent biological replicates (stated in figure legends) were quantified and compared using GraphPad Prism. For the SQLE stability assay, the median of the mCherry:GFP ratio of the mutants were compared to the WT ratio using one-way ANOVA. $P$ values and significance summary for each mutant are given in the Supplementary Data 3. Comparison of mRNA levels for low abundant mutants was also done using ANOVA and significance levels are highlighted above each mutant bar. For the SQLE-binding assay, the WT ratio of SQLE to MARCH6-FLAG was compared to the respective mutant ratios using a two-tailed $t$ test. Significance levels in graphs are given for the calculated $P$ values: 0.1234 (ns), 0.0332 (*), 0.0021 (**), < 0.0002 (***).

## Reporting summary

Further information on research design is available in the Nature Portfolio Reporting Summary linked to this article.

## Data availability

The structural data will be available from EMDB and RCSB upon manuscript publication. Doa10 in MSP1E3D1 without the RING domain: EMDB-17597, PDB: 8PD0; Doa10 in MSP1E3D1 with the RING domain: PDB: 8PDA. Low resolution cryo-EM maps were deposited to EMDB and are available via the following accession numbers: Doa10 complex in MSP1E3D1: EMDB-17609, Doa10 in MSP2N2: EMDB-17610. Mass spectrometry data were deposited in the ProteomeXchange Consortium with dataset identifier PXD047499. Raw gel images, western blots and raw data underlying plots are provided as Source Data or Supplementary Figures. Source data are provided with this paper.

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

## Acknowledgements

We thank B. Steigenberger and the mass spectrometry facility at the Max Planck Institute of Biochemistry for help with mass spectrometry; D. Bollschweiler and T. Schäfer and the cryo-EM facility at the Max Planck Institute of Biochemistry for assistance with cryo-EM; J. Kellermann, C. Baumann and A. Alpi for administrative and technical assistance; J. R. Prabu, K. Baek and L. Hopf for help with cryo-EM processing and structure building; D. Wöhlert, W. Kühlbrandt, M. Liao, T. Raisch and S. Raunser for advice on cryo-EM sample preparation and data processing; J. Weissman, G. Muthukumar, A. Xu, L. Miller-Vedam and the entire Weissman laboratory for reagents, advice and protocols for CRISPRi and fluorescent reporter assay; A. Guna for the K562 UCOE-SFFV-Zim3-dCas9-P2A-hygro cell line prior to publication; R. Farese and E. Conti for helpful discussions; M. Oster and M. Spitaler and the imaging facility at the Max Planck Institute of Biochemistry for assistance with flow cytometry; A. Mehrtasch and M. Hochstrasser for sharing yeast strains and plasmids; L. Bas, V. Beier and C. Langlois for assistance in yeast genetics. This work was funded by the Max Planck Society, the European Union (ERC AdvG, UPSmeetMet, 101098161 to BAS; HORIZON-MSCA, Redox in macrophages, 101062335 to PPO), and the Deutsche Forschungsgemeinschaft (DFG BR 6742/1-1, Molecular Mechanisms of Membrane Protein Quality Control to BB).

## Author contributions

Conceptualization: B.B., J.J.B. and B.A.S.; Cryo-EM and structure building: J.J.B.; Molecular biology: J.J.B., B.B., L.S.; Protein purification and expression: J.J.B., R.J. and S.v.G.; Sybody selection: J.J.B., M.S., M.A.S.; in vitro assays: J.J.B.; Cell-based assays: J.J.B. and B.B.; qPCR: B.B. and P.P.O. (supervised by P.J.M.); Manuscript preparation: B.B., J.J.B, B.A.S. and P.J.M., with input from all authors.

## Funding

## Competing interests

B.A.S. is adjunct faculty at St. Jude Children's Research Hospital, Memphis, TN, USA, is on the Scientific Advisory Boards of BioTheryX and Proxygen, and is co-inventor of intellectual property related to DCN1 inhibitors (unrelated to this work) licensed to Cinsano. M.A.S. is co-founder and shareholder of Linkster Therapeutics AG. The remaining authors declare no competing interest.
