## [Peer Review File · Nature Communications]

Doa10/MARCH6 architecture interconnects E3 ligase activity with lipid-binding transmembrane channel to regulate SQLEREVIEWER COMMENTS

Reviewer #1 (Remarks to the Author):

This manuscript focuses on the ER E3 ligase yeast Doa10/human MARCH6, with noteworthy results. Structural analyses are done by solving a cryo-EM structure for Doa10 and using AlphaFold2 and AlphaFold-Multimer respectively to predict the structure of MARCH6 and its complex with Ube2J2 and ubiquitin. The MARCH6 predicted structure is similar to the Doa10 experimental structure and an interesting feature is that the authors find lipid-binding sites in Doa10. The structural analyses are complemented with a fluorescence-based activity assay that monitors the MARCH6 substrate SQLE, with 96 MARCH6 mutations evaluated. A curiosity that should probably be addressed is why the authors didn't solve the structure of MARCH6, most likely due to technical challenges but it'd be worth mentioning this in the manuscript. The findings are of significance to the field and the following suggestions may help clarify points and increase confidence in the conclusions.

1. Figure 1 and S1 convincingly demonstrate that the reporter assay can measure MARCH6 activity in cells. A minor suggestion is to define M6 as MARCH6 in the text and caption.

2. Figure 2: A domain layout with amino acid numbers should be included with this figure. The text notes that amino acids 240 – 468 are not visible but without explicitly stating where this region is relative to the functional/structural domains. Similarly, a figure showing the lipid bilayer mid-plane definition relative to the RING domain prediction is needed. This information could be included in the right image of Fig. 2A. Also, an image showing the local resolution throughout the structure is needed – this could be included with Supplementary Figure 2.

3. p. 6 and Supplementary Fig. 4: 'expression levels' is probably not the best descriptor here and below, as the assay measures 'protein abundance' and in this case reduced levels may be due to protein instability/auto-ubiquitylation. For the few mutants with lower levels, it would be worth doing qPCR to test whether protein expression (mRNA) is affected and treatment with proteasome and/or E1 inhibitors to test for rescue.

A minor suggestion for Supplementary Fig. 4 is to use indicators for each blot (i), (ii), ... This would help readers to find the specific mutations referred to in the text.

4. Figure 3: It's not clear how confidently the RING position is defined. This could be addressed by the local resolution map suggested in #2.

5. Figure 4: By using AF2-Multimer, the binding site of Ube2J2 and ubiquitin are predicted on MARCH6. The prediction is supported by the activity assay. However, the authors should also directly test for impaired interactions for the supporting mutants (for example MARCH6 E5A/E6A/E15A/E19A). This could be done for example by expressing tagged MARCH6 WT or mutated protein and using immunoprecipitation experiments or by using recombinant proteins.

6. Figure 5: Similar to the comment above, the manuscript would be stronger if the authors directly tested whether SQLE interaction with MARCH6 is reduced by the mutations in the membrane channel.

7. p. 9 – 10: The authors write "Alternatively, the channel may be a site of SQLE engagement." This sentence is duplicative with a sentence before it and it's not clear what the authors mean by the alternative models.

8. Figure 6: The interactions with the phospholipids is interesting and the amino acid mutations in this region supportive of a functional role. It would be worth doing a functional assay for Doa10 for these mutations since the structural information is for the yeast protein. Related to comment #2 above, including information on the local resolution in these lipid-binding regions would be informative. Also, are there other mutations in the lipid-binding regions? It'd be worth integrating these results, with inclusion of cases that do not affect SQLE levels.

Reviewer #2 (Remarks to the Author):

Overview

In this manuscript, the authors present structural, predicted, and biochemical evidence describing the molecular functionality of the ER-resident ubiquitin ligase family that includes Doa10/MARCH6. They employ a fluorescent SQLE reporter system which is used to test an extensive array of MARCH6 mutant deduced from a low-resolution Cryo-EM structure of Doa10 augmented with AlphaFold (AF) predictions. In Figure 1, the authors describe their ratiometric SQLE reporter – which is stabilised in K562 cells knocked down for MARCH6 by CRISPRi and is degraded upon reintroduction of an MARCH6 rescue plasmid but not one which lacks the RING domain or contains the G885L mutation. It also responds to increases in exogenously added cholesterol. And although near the RING domain, a C-term tag did not affect its activity toward SQLE (Supp Fig 1). In Figure 2, the authors used Cryo-EM to determine structures for Doa10 (the MARCH6 yeast orthologue) bound to Ubc6 (E2) at 3.58Å. The structure identified all TMs and report a 4 TM bundle referred to the flexible gate – suggested by potential movement. A central channel was visible as was the RING domain. This structure and the AF model were comparable and suggest that the MARCH6 AF model would be as representative (Sup Fig. 2,3). Based on these structures, the authors devised a structure guided mutagenesis effort for MARCH6 to identify key domains/regions required for SQLE degradation. Testing 197 mutations, the authors identified 96 that participated in function as tested. Only several were unstable and some even stabilised MARCH6. Testing SQLE – mutants were found which impaired turnover as well as ones that enhanced it (gain of function, Q866A/Q869A). These were present throughout the structure in different domains (e.g channel lining, RING proximal, lipid facing, etc). Similar observations were found for Plin2 (Supp Fig 5) but mostly for strongest mutants. The authors next focused on the area encompassing the RING domain of MARCH6, modelling it with Ube2j2 and identifying and screening various mutants that impact RING positioning close to the E2 and Ub (Figure 4). Another observation reported is the RING position above the central channel through a beta-sheet structure. Mutations in this beta sheet disrupt SQLE degradation and cause MARCH6 to accumulate. These data argue for important roles in the RING flanking regions. The authors next highlight a wide channel predicted by the structures and differential positioning for the gate regions, speculating the capacity to surround a TMD and side chains (Figure 5). Contrasting data for SQLE and PLIN2 stabilisation led the authors to suggest the channel adjusts for different substrates – through channels dimensions and the lining properties. To conclude, the authors suggest lipid binding sites that are conserved for which mutations cause defects in SQLE degradation (Figure 6). The authors summarise with a model for SQLE engagement and nice comparison with other structures of related membrane-bound ubiquitylation machinery.

Overall impression

Structural and functional insight into membrane embedded ubiquitin ligases participating in ERAD is a welcome addition to the plethora of genetic and biochemical data that has defined our understanding for some time. It should be of general interest both in its strategy and in its findings. This manuscript does a very good job of executing this herculean study of different mutants and in doing so, further refining our understanding of key interactions as well as structural parameters that influence function. Overall, the manuscript is well written, the experiments performed to a high standard, the data of good quality, and the appropriate statistical analyses have been included where necessary and are described. The Methods and Materials describe the experiments that have been performed. Some concern might arise from the fact that some of the direct evidence from a MARCH6 structure may be lacking, but the conservation of many domains/residues from Doa10 has helped to assuage this reviewer's concern. The substrate interactions with a channel region are a bit more speculative and the study would have benefited from cross-linking assays that have been used to define substrate contact domains in the yeast Hrd1. Nevertheless, such experiments are beyond the scope of this paper and so should not hold up these studies, which are extensive in their own right. The manuscript is an advancement of understanding of the MARCH6/Doa10 system for ERAD. There are just a few

comments/suggestions that this reviewer has regarding the data presented. They are listed below.

Queries & Comments

1. In Figure 1c, for clarity please indicate the exact mutant in the lower panel (MARCH6-G885L)
2. Could the authors address why in some instances multiple mutations were required to see effects? Was their design intentional or a consequence of the strategy to generate mutations.
3. Figure 3b maps the SQLE stabilisation by predicted domain. These appear to be average values, so could the authors please include error bars.

Reviewer #3 (Remarks to the Author):

In the current report, Botsch et al. (2023) describe the development of an assay system to track human MARCH6 activity and determine the structure of yeast Doa10. The study is based on a substantial body of mutational data and provides valuable insights into the structure-guided mechanism of MARCH6 E3 ligase activity. Understanding the molecular architecture of Doa10 is particularly significant due to its implications for the quality control mechanism of ER-targeting proteins in ERAD. However, before considering publication in Nature Communications, the following comments should be addressed.

Major issues:

1. The extensive description of the functional aspects of Doa10 structure heavily relies on mutational data, specifically regarding the stability of the MARCH6 substrate (SQLE). While these mutagenic experiments yield informative observations, it is important to acknowledge their limitations in revealing the structural aspects of the mutation sites. The deficiency of SQLE degradation could be attributed to multiple factors. To propose structural models, it is essential to equally consider alternative possibilities. For example, if potential substrate binding sites are mutated and SQLE degradation is disrupted, measuring their binding affinity could further support the proposed model. Accompanying experiments such as isothermal titration calorimetry or surface plasmon resonance should be conducted to firmly establish these aspects. Without this additional support, the main findings and mechanisms cannot be adequately justified. It is recommended to revisit each results section and provide appropriate experimental validations.
2. Although Ubc6 and the sybody were present in the cryo-EM sample, no extra densities corresponding to them were observed. This is unfortunate since understanding the association of Ubc6 with Doa10 could provide significant insights into how Doa10 facilitates the transfer of Ub from Ubc6 to its substrates. Given that Ubc6 and Doa10 were co-eluted during gel filtration, it is plausible that they are bound even on the grid. Supplementary Fig. 2 lacks 3D classification, which is often necessary to obtain additional maps for weakly-bound or conformationally flexible proteins. Therefore, it is recommended to perform extensive 3D classification to potentially visualize the Ubc6 map and improve the map resolution.
3. While the molecular architecture of Doa10 itself provides valuable information, several important hypotheses put forth by the authors rely on the docking model by AF multimer (Fig. 4). Although the mutational data is appreciated, it is hypothesized that MARCH6, Ube2J2, and Ub bind at the exact binding interface as suggested by the docking model. Thus, it is crucial to conduct further binding experiments using recombinant proteins to validate the proposed model.
4. The models described in the second and third paragraphs of the discussion are intriguing; however, they remain highly speculative. To persuade readers, it is necessary to provide supporting experimental results from either the authors or other relevant literature.
5. The interpretation of lipid-binding sites within the scaffold domain is based on mutational results of residues coordinated with lipids. The authors speculate that cholesterol might bind to these

specific sites due to MARCH6's role in regulating cholesterol homeostasis. However, this speculation is not sufficiently supported. Cholesterol density can usually be distinguished from that of phospholipids, and intercalated phospholipids often stabilize the overall architecture of membrane proteins. Therefore, the hypothesis regarding cholesterol binding requires more evidence.

6. Considering the relatively large deviation of the median mCherry:GFP ratios for each MARCH6 variant, the number of repeats is insufficient, especially for critical mutants that support the main results of the manuscript. It is recommended to increase the number of repeats for critical mutants and provide p-values.

Minor comments:

1. The observation that extra densities are distinct depending on the size of nanodiscs (Fig. 2C) is interesting. It suggests that the flexibility of additional maps can be modulated by nanodisc size. It would be worthwhile to explore other MSP proteins with a wider range of size distributions to visualize Ubc6.

2. Conducting 3D variability analysis would be useful to assess the structural flexibility of the gate (Fig. 7).

3. The discussion section should be thoroughly rewritten to minimize speculation and consider alternative explanations supported by previous findings in the literature.

REVIEWER COMMENTS

Reviewer #1 (Remarks to the Author):

This manuscript focuses on the ER E3 ligase yeast Doa10/human MARCH6, with noteworthy results. Structural analyses are done by solving a cryo-EM structure for Doa10 and using AlphaFold2 and AlphaFold-Multimer respectively to predict the structure of MARCH6 and its complex with Ube2J2 and ubiquitin. The MARCH6 predicted structure is similar to the Doa10 experimental structure and an interesting feature is that the authors find lipid-binding sites in Doa10. The structural analyses are complemented with a fluorescence-based activity assay that monitors the MARCH6 substrate SQLE, with 96 MARCH6 mutations evaluated. A curiosity that should probably be addressed is why the authors didn't solve the structure of MARCH6, most likely due to technical challenges but it'd be worth mentioning this in the manuscript. The findings are of significance to the field and the following suggestions may help clarify points and increase confidence in the conclusions.

We thank the reviewer for the positive evaluation of our manuscript. Regarding human MARCH6, we indeed tried extensively to obtain its structure alone or in complex with Ube2J2. However, even when expressed in mammalian cells and purified in mild detergent, the recombinant protein behaved poorly (very broad SEC profile, anomalous behavior in thermal denaturation assays) and we could not obtain interpretable cryo-EM reconstructions.

We have modified the relevant parts in the manuscript and legends to reflect these changes:

“Although our cryo-EM sample contained Ubc6 and the sybody, we did not observe unambiguous density for either, even after extensive 3D classification (Supplementary Fig. 2b). Efforts to obtain cryo-EM structures of human MARCH6, alone or with Ube2J2, in different reconstitution systems, remained unsuccessful.”

1. Figure 1 and S1 convincingly demonstrate that the reporter assay can measure MARCH6 activity in cells. A minor suggestion is to define M6 as MARCH6 in the text and caption.

We now consistently refer to the protein as MARCH6 throughout all text and figures.

2. Figure 2: A domain layout with amino acid numbers should be included with this figure. The text notes that amino acids 240 – 468 are not visible but without explicitly stating where this region is relative to the functional/structural domains. Similarly, a figure showing the lipid bilayer mid-plane definition relative to the RING domain prediction is needed. This information could be included in the right image of Fig. 2A. Also, an image showing the local resolution throughout the structure is needed – this could be included with Supplementary Figure 2.

We have now moved the Doa10/MARCH6 domain structure figure (originally Supplementary Figure 3b) to the top of Figure 2 (new Figure 2a). Figure 2b (formerly Figure 2a) now marks the approximate lipid bilayer mid-plane and its relative distance to the RING domain. Additionally, we have included local resolution maps in Supplementary Figure 3 (formerly Supplementary Figure 7b).

We have modified the relevant parts in the manuscript and legends to reflect these changes:

“The cytosolic region between residues 240-468, predicted to be unstructured (Fig. 2a), was not visible in the map. Examining the map at lower threshold showed a bundle of four transmembrane helices (Helices 12-15, residues 1108-1303) (Fig. 2a). This area, which we call the flexible gate, had significant differences in the complex reconstituted in narrow and wide nanodiscs, seen in our medium-resolution cryo-EM data (Fig. 2c, Supplementary Fig. 5a, b), indicating movement in this

protein part. Flexibility of this region is also suggested by the poorer local map resolution of gate helices compared to the scaffold (Supplementary Fig. 3a)”

3. p. 6 and Supplementary Fig. 4: ‘expression levels’ is probably not the best descriptor here and below, as the assay measures ‘protein abundance’ and in this case reduced levels may be due to protein instability/auto-ubiquitylation. For the few mutants with lower levels, it would be worth doing qPCR to test whether protein expression (mRNA) is affected and treatment with proteasome and/or E1 inhibitors to test for rescue.

We now use “protein abundance” instead of “expression levels” in the text to make this distinction. In addition, we have i) performed qPCR to measure mRNA levels of several MARCH6 mutants where we have observed lower or higher protein abundance compared to WT (W274A, W292L, H605A, E910A in addition to Δ RING), ii) performed FLAG-immunoblot of these mutants with or without proteasome inhibition. The qPCR data is now included in a new Supplementary Fig. 6 subpanel (Supplementary Fig. 6b). With qPCR, we see no statistically significant difference in mRNA abundance between WT March6 and any of the mutants tested, suggesting indeed that any major differences in protein abundance between these variants are likely due to post-translational protein stabilities. When blotting for the protein abundance of these mutants in presence or absence of proteasome inhibition, we see no strong accumulation of these mutants when proteasome is inhibited by MG132. At this point we would prefer to not interpret the proteasome inhibition results and chose to leave the data out of the manuscript. The pathways regulating MARCH6 turnover (proteasome and/or autophagy) require future study beyond the scope of the current work.

Reviewer response Figure 1: Immunoblot against MARCH6 variants with or without proteasome inhibition.

We want to also point out that we removed the MARCH6^{W93A/I862A} mutant from our study: this mutant was nearly as defective in SQLE regulation as MARCH6 ^{Δ RING} but was not expressed. In re-sequencing the expression plasmids, we found a frameshift mutation that would lead to a premature translation termination.

We have modified the relevant parts in the manuscript and legends to reflect these changes:

“As a quality control readout, we also examined protein abundance of each mutant by anti-FLAG immunoblotting, alongside MARCH6^{WT} and MARCH6 ^{Δ RING} as benchmarks (Supplementary Fig. 6a). This analysis revealed that only a few mutants were less abundant compared to MARCH6^{WT}, suggesting that the protein fold was retained in the majority of variants studied. Additional qPCR analysis of a subset of lower-expressing mutants confirmed that mRNA levels were comparable to WT levels. Thus, we surmised that lower protein abundance in some mutants likely resulted from post-translational instability (Supplementary Fig. 6b)”

A minor suggestion for Supplementary Fig. 4 is to use indicators for each blot (i), (ii), ... This would help readers to find the specific mutations referred to in the text.

We have added additional indicators to the immunoblot panels in Supplementary Fig. 6a to make finding each mutant easier for the reader. Our Supplementary Table 2 summarizing SQLE reporter effects for each mutant now includes these indications to aid the reader in finding the respective immunoblot data.

4. Figure 3: It's not clear how confidently the RING position is defined. This could be addressed by the local resolution map suggested in #2.

The RING density has been included in the local resolution map in Supplementary Figure 3b as suggested.

5. Figure 4: By using AF2-Multimer, the binding site of Ube2J2 and ubiquitin are predicted on MARCH6. The prediction is supported by the activity assay. However, the authors should also directly test for impaired interactions for the supporting mutants (for example MARCH6 E5A/E6A/E15A/E19A). This could be done for example by expressing tagged MARCH6 WT or mutated protein and using immunoprecipitation experiments or by using recombinant proteins.

We thank the reviewer for raising this important point. To address the validity of the AF multimer predicted interfaces between the catalytic, cytosolic domains of MARCH6 and Ube2J2 with ubiquitin (Figure 4a), we have performed E2~ubiquitin discharge assays. We included mutants in the MARCH6 RING domain, the Ube2J2 catalytic domain and ubiquitin, designed to weaken the predicted interactions between the three proteins. The discharge assay is an established functional assay to verify the formation of a ternary RING-E2~ubiquitin complex, in which the RING domain stabilizes the closed E2~ubiquitin conformation to promote discharge of the ubiquitin from the E2~ubiquitin complex (Saha et al., 2011, Mol. Cell.).

We first mutated the predicted interface between the MARCH6 RING and the Ube2J2 catalytic domain (Supplementary Fig. 8a), which resembles previously observed RING-E2 interactions, as exemplified by the RING-UBCH5B~ubiquitin structures (Plechanovová et al., 2012, Nature; Dou et al., 2012, Nat. Struct. Mol. Biol.). Here, mutating the conserved MARCH6 V11 to either V11A or V11D caused moderate to severe discharge defects, respectively (Fig. 4b, Supplementary Fig. 8b), consistent with the hydrophobic interactions between these domains.

Next, we mutated the predicted Ube2J2-ubiquitin interface, centered on the ubiquitin I44 patch (Supplementary Fig. 8a). Here, mutating either Ube2J2 S120 or T116 to bulky residues caused strong discharge defects, which could be partially rescued when using a reciprocal ubiquitin I44A mutant (Fig. 4c, Supplementary Fig. 8c). Mutation of these two Ube2J2 residues to a smaller glycine resulted in discharge defects which could not be rescued by ubiquitin I44A (Supplementary Fig. 8c).

Together, our mutational analysis of the RING-E2 and E2-ubiquitin interfaces support the AF model for a ternary MARCH6(RING)-Ube2J2-ubiquitin catalytic complex.

Furthermore, we used this discharge assay to test the MARCH6^{E5A/E6A/E15A/E19A} mutation, which caused a strong defect in our SQLE stability assay. We suggested a catalytic role for this mutant based on the close proximity to the RING domain, the AF model of the catalytic moieties and the increase in protein abundance (similar to other catalytic defective mutants like Δ RING). Indeed, the mutation also results in a strong discharge defect (Fig. 4b) further supporting a possible role in stabilizing the catalytic domains for ubiquitin discharge.

We have rewritten the relevant part of the Results section to incorporate these new findings:

“For structural insights into mutants suggested to play a catalytic role, we used AF-multimer⁵² to model a MARCH6-ubiquitin-Ube2J2 complex for both the full-length protein complex, spanning the membrane (Fig. 4a), and only the soluble catalytic domains focusing on the ubiquitination active site (Supplementary Fig. 8a). Despite close resemblance to known RING-E2~ubiquitin structures (Supplementary Fig. 8a), we tested the predicted positioning of the catalytic moieties using an ubiquitin discharge assay^{53, 54}. This assay follows the release of ubiquitin from the thioester bond with Ube2J2, stimulated by interactions between E2~ubiquitin and the RING domain. Indeed, adding WT MARCH6 RING to the loaded Ube2J2~ubiquitin complex accelerates ubiquitin discharge in our in vitro assay (Fig. 4b and Supplementary Fig. 8b).

To further assess the positioning of the catalytic domains, we mutated conserved residues predicted to be localized at the interfaces between these domains (Supplementary Fig. 8a). First, we mutated MARCH6 RING V11 predicted to interact with a hydrophobic patch in UBE2J2 (Supplementary Fig. 8a). Consistent with the model, mutations to aspartic acid or alanine lead to a strong or intermediate defect, respectively, in RING-mediated ubiquitin discharge from Ube2J2~ubiquitin (Fig. 4b, Supplementary Fig. 8b). Next, we targeted the predicted interface between the E2 catalytic domain with the ubiquitin I44 patch (Fig. 4c, Supplementary Fig. 8a, c, d). Introducing bulky residues at this interface (S120F, S120R, T116F, T116R) results in impaired ubiquitin discharge with WT ubiquitin. These discharge defects are partially rescued with a ubiquitin I44A mutation designed to compensate for the larger opposing E2 residues (Fig. 4c, Supplementary Fig. 8c). Furthermore, E2 mutations to glycine have milder effects, which are not rescued by ubiquitin I44A in agreement with the model (Supplementary Fig. 8c).

In our broad mutagenesis campaign, we found mutations in the MARCH6 N-terminus (MARCH6^{E5A/E6A/E15A/E19A}) highly defective in mediating SQLE degradation (Fig. 3b, cyan). Our MARCH6-Ube2J2-ubiquitin AF model predicts MARCH6 E5 to form a bond with Ube2J2 residue K26 (Fig. 4d), suggesting additional catalytic domain stabilization by this N-terminus. Indeed, the isolated MARCH6^{E5A/E6A/E15A/E19A} RING domain has a defect in stimulating ubiquitin discharge from Ube2J2 compared to WT (Fig. 4b), supporting a role of the N-terminus. Finally, we also observed accumulation of MARCH6^{E5A/E6A/E15A/E19A} protein, emphasizing the importance of the MARCH6 N-terminus for catalysis, including auto-degradation (Supplementary Fig. 6a).”

6. Figure 5: Similar to the comment above, the manuscript would be stronger if the authors directly tested whether SQLE interaction with MARCH6 is reduced by the mutations in the membrane channel.

We have performed immunoprecipitation (IP)-Western blot (WB) experiments to assess interactions between re-expressed MARCH6-FLAG variants with endogenous SQLE in K562 cell lines (Figure 5d). We selected MARCH6 membrane channel lining mutants displaying defects in our SQLE reporter screen: Y92V, Y96V, L140A/L747A and Y92V/Y96V. Indeed, we find that MARCH6^{Y96V} and MARCH6^{Y92V/Y96V} show markedly weakened interactions with SQLE compared to WT. Together with our total proteomics data on the Y96V mutant (Supplementary Fig. 11b), these results are consistent with MARCH6 engaging SQLE's N-terminal helices within the membrane channel.

While the precise mode of SQLE binding inside the membrane channel remains to be verified experimentally – also in regard to what function the channel lining residues identified in our screen play exactly – the AF model of the E3-substrate complex, along with our IP-WB and proteomics data, indeed support the model in which the MARCH6 membrane channel is a likely site of substrate engagement, above which the E3 architecture positions an active ubiquitination active site in the cytoplasm.

We have modified the relevant parts in the manuscript and legends to reflect these changes:

“The model suggests that mutations lining the channel could affect direct binding and/or orientation of the substrate relative to the ubiquitylation active site. We thus surveyed these mutants by performing FLAG-immunoprecipitation for cell lines expressing FLAG-tagged MARCH6 variants followed by Western blotting against endogenously bound SQLE (Fig. 5d). This revealed a range of effects. The decreased amounts of endogenous SQLE co-precipitating with MARCH6^{Y96V} and MARCH6^{Y92V/Y96V} compared to MARCH6^{WT}, suggest that defective SQLE degradation in the Y96V mutant background can in part be explained by weakened interaction between SQLE and this region of the MARCH6 membrane channel. Meanwhile, the other mutations maintained interaction and thus may affect capacity for SQLE ubiquitylation and turnover.”

7. p. 9 – 10: The authors write “Alternatively, the channel may be a site of SQLE engagement.” This sentence is duplicative with a sentence before it and it’s not clear what the authors mean by the alternative models.

We have entirely re-written this part of the text for clarity:

“Several MARCH6 mutants, exhibiting both defective and GOF phenotypes, are predicted to line the membrane channel interior (Fig. 3b, magenta). We hypothesized that the membrane channel could be a site for SQLE engagement, based on mutational effects, and its varying width, hydrophobicity, and location relative to the RING domain (Supplementary Fig. 10a, b). We modeled both the complex with full-length SQLE, or with its N-terminal 100 residues (SQLE-N100). SQLE-N100 was previously shown to be sufficient for recognition and ubiquitin-dependent degradation by MARCH6^{30, 31}. SQLE's N-terminal helices were placed inside MARCH6's membrane channel, although the models differed in orientation of the two SQLE helices (Fig. 5c). Defective MARCH6 mutants predicted to be near the SQLE N-terminus include MARCH6^{Y92V}, MARCH6^{Y96V}, and MARCH6^{L140A/L747A}. The strongest of our GOF mutants, MARCH6^{Q866A/Q869A}, is also predicted to be close to the bound substrate.”

8. Figure 6: The interactions with the phospholipids is interesting and the amino acid mutations in this region supportive of a functional role. It would be worth doing a functional assay for Doa10 for these mutations since the structural information is for the yeast protein. Related to comment #2 above, including information on the local resolution in these lipid-binding regions would be informative. Also, are there other mutations in the lipid-binding regions? It’d be worth integrating these results, with inclusion of cases that do not affect SQLE levels.

We have updated Fig. 6 to include an additional MARCH6 residue we mutated based on the Doa10 lipid-proximal residues (Fig. 6d). This residue, K374 in MARCH6 and K625 in Doa10, is interacting with a bound phospholipid (Lipid 3) in the yeast homolog. We show that mutation of MARCH6 K374 to either alanine or glutamate has no apparent effect on SQLE reporter levels. This is in contrast to the observed effects of mutating MARCH6 K116 or F453, where homologous residues in Doa10 are involved in binding Lipid 1 and Lipid 2, respectively.

We have also included local resolution map for the Doa10 lipid binding sites in the updated Supplementary Fig. 3c.

We have modified the relevant parts in the manuscript to reflect these changes:

“In addition to these two positionally conserved lipid binding sites just described, we mutated a third potential MARCH6 scaffold lipid binding site (Lipid 3), where we observe an interaction between Doa10 K625 and the lipid phosphate group (Fig. 6d). Unlike the Lipid 1 and Lipid 2 sites, neither MARCH6^{K374A} nor MARCH6^{K374E} showed significant SQLE degradation defects.”

Reviewer #2 (Remarks to the Author):

Overview

In this manuscript, the authors present structural, predicted, and biochemical evidence describing the molecular functionality of the ER-resident ubiquitin ligase family that includes Doa10/MARCH6. They employ a fluorescent SQLE reporter system which is used to test an extensive array of MARCH6 mutant deduced from a low-resolution Cryo-EM structure of Doa10 augmented with AlphaFold (AF) predictions. In Figure 1, the authors describe their ratiometric SQLE reporter – which is stabilised in K562 cells knocked down for MARCH6 by CRISPRi and is degraded upon reintroduction of an MARCH6 rescue plasmid but not one which lacks the RING domain or contains the G885L mutation. It also responds to increases in exogenously added cholesterol. And although near the RING domain, a C-term tag did not affect its activity toward SQLE (Supp Fig 1). In Figure 2, the authors used Cryo-EM to determine structures for Doa10 (the MARCH6 yeast orthologue) bound to Ubc6 (E2) at 3.58Å. The structure identified all TMs and report a 4 TM bundle referred to the flexible gate – suggested by potential movement. A central channel was visible as was the RING domain. This structure and the AF model were comparable and suggest that the MARCH6 AF model would be as representative (Supp Fig. 2,3). Based on these structures, the authors devised a structure guided mutagenesis effort for MARCH6 to identify key domains/regions required for SQLE degradation. Testing 197 mutations, the authors identified 96 that participated in function as tested. Only several were unstable and some even stabilised MARCH6. Testing SQLE – mutants were found which impaired turnover as well as ones that enhanced it (gain of function, Q866A/Q869A). These were present throughout the structure in different domains (e.g channel lining, RING proximal, lipid facing, etc). Similar observations were found for Plin2 (Supp Fig 5) but mostly for strongest mutants. The authors next focused on the area encompassing the RING domain of MARCH6, modelling it with Ube2j2 and identifying and screening various mutants that impact RING positioning close to the E2 and Ub (Figure 4). Another observation reported is the RING position above the central channel through a beta-sheet structure. Mutations in this beta sheet disrupt SQLE degradation and cause MARCH6 to accumulate. These data argue for important roles in the RING flanking regions. The authors next highlight a wide channel predicted by the structures and differential positioning for the gate regions, speculating the capacity to surround a TMD and side chains (Figure 5). Contrasting data for SQLE and PLIN2 stabilisation led the authors to suggest the channel adjusts for different substrates – through channels dimensions and the lining properties. To conclude, the authors suggest lipid binding sites that are conserved for which mutations cause defects in SQLE degradation (Figure 6). The authors summarise with a model for SQLE engagement and nice comparison with other structures of related membrane-bound ubiquitylation machinery.

Overall impression

Structural and functional insight into membrane embedded ubiquitin ligases participating in ERAD is a welcome addition to the plethora of genetic and biochemical data that has defined our understanding for some time. It should be of general interest both in its strategy and in its findings. This manuscript does a very good job of executing this herculean study of different mutants and in doing so, further refining our understanding of key interactions as well as structural parameters that influence function. Overall, the manuscript is well written, the experiments performed to a high standard, the data of good quality, and the appropriate statistical analyses have been included where necessary and are described. The Methods and Materials describe the experiments that have been performed. Some concern might arise from the fact that some of the direct evidence from a MARCH6 structure may be lacking, but the conservation of many domains/residues from Doa10 has helped to assuage this reviewer's concern. The substrate interactions with a channel region are a bit more speculative and the study would have benefited from cross-linking assays that have been used to define substrate contact domains in the yeast Hrd1. Nevertheless, such experiments are beyond the scope of this paper and so should not hold up these studies, which are extensive in their own right. The manuscript is an advancement of understanding of the MARCH6/Doa10 system for ERAD. There

are just a few comments/suggestions that this reviewer has regarding the data presented. They are listed below.

We thank the reviewer for the positive evaluation of our manuscript.

Queries & Comments

1. In Figure 1c, for clarity please indicate the exact mutant in the lower panel (MARCH6-G885L)

We have modified Figure 1c accordingly.

2. Could the authors address why in some instances multiple mutations were required to see effects? Was their design intentional or a consequence of the strategy to generate mutations.

When designing mutants based on the predicted MARCH6 AF structure, we often looked to mutate groups of residues which appeared to form “patches”. This was done both for practical reasons (allowing us to screen approximately twice the number of residues as the number of ligase mutants) and also to increase the likelihood of observing significant mutant effects. We reasoned that especially for hydrophobic patches inside the lipid bilayer, mutating more than one residue would likely be required to markedly perturb binding interfaces in order to produce measurable effects using our reporter assay.

3. Figure 3b maps the SQLE stabilisation by predicted domain. These appear to be average values, so could the authors please include error bars.

To address this suggestion and a related comment from Reviewer 3, we have included additional replicates for each mutant in the revised manuscript (new Figure 3b). We now also show error bars (S.E.M.) along with the individual data points. In our Supplementary Table 2, which summarizes all mutant effects, we have included a column with *P* values derived from an ANOVA pairwise comparing the SQLE reporter levels for each mutant against MARCH6 WT.

Reviewer #3 (Remarks to the Author):

In the current report, Botsch et al. (2023) describe the development of an assay system to track human MARCH6 activity and determine the structure of yeast Doa10. The study is based on a substantial body of mutational data and provides valuable insights into the structure-guided mechanism of MARCH6 E3 ligase activity. Understanding the molecular architecture of Doa10 is particularly significant due to its implications for the quality control mechanism of ER-targeting proteins in ERAD. However, before considering publication in Nature Communications, the following comments should be addressed.

Major issues:

1. The extensive description of the functional aspects of Doa10 structure heavily relies on mutational data, specifically regarding the stability of the MARCH6 substrate (SQLE). While these mutagenic experiments yield informative observations, it is important to acknowledge their limitations in revealing the structural aspects of the mutation sites. The deficiency of SQLE degradation could be attributed to multiple factors. To propose structural models, it is essential to equally consider alternative possibilities. For example, if potential substrate binding sites are mutated and SQLE degradation is disrupted, measuring their binding affinity could further support the proposed model. Accompanying experiments such as isothermal titration calorimetry or surface plasmon resonance should be conducted to firmly establish these aspects. Without this additional support, the main findings and mechanisms cannot be adequately justified. It is recommended to revisit each results section and provide appropriate experimental validations.

We thank the reviewer for bringing up this important point. Indeed, while our fluorescent assay reports on SQLE reporter stability following re-introduction with mutant MARCH6, it is not a direct experimental demonstration of loss-of-binding in cases where mutants were designed based on predicted ligase-substrate interfaces. We would like to point out that ITC or SPR, as the reviewer suggests, are not standard approaches for studying interactions of two membrane proteins within a native lipid bilayer. Thus, in response to this comment and a related suggestion from Reviewer #1, as an alternative approach to address this issue, we performed immunoprecipitation (IP)-Western blotting (WB) to assess binding of channel lining MARCH6 mutants to SQLE (Figure 5d).

As detailed in our above response to Reviewer #1 Point 6, we selected MARCH6 membrane channel lining mutants displaying defects in our SQLE reporter screen: Y92V, Y96V, L140A/L747A and Y92V/Y96V. Indeed, we find that MARCH6^{Y96V} and MARCH6^{Y92V/Y96V} show markedly weakened interactions with SQLE compared to WT. Together with our total proteomics data on the Y96V mutant (Supplementary Fig. 11b), these results suggest that MARCH6 likely engages SQLE's N-terminal helices within the membrane channel.

We have modified the relevant parts in the manuscript to reflect these changes:

“The model suggests that mutations lining the channel could affect direct binding and/or orientation of the substrate relative to the ubiquitylation active site. We thus surveyed these mutants by performing FLAG-immunoprecipitation for cell lines expressing FLAG-tagged MARCH6 variants followed by Western blotting against endogenously bound SQLE (Fig. 5d). This revealed a range of effects. The decreased amounts of endogenous SQLE co-precipitating with MARCH6^{Y96V} and MARCH6^{Y92V/Y96V} compared to MARCH6^{WT}, suggest that defective SQLE degradation in the Y96V mutant background can in part be explained by weakened interaction between SQLE and this region of the MARCH6 membrane channel. Meanwhile, the other mutations maintained interaction and thus may affect capacity for SQLE ubiquitylation and turnover.”

2. Although Ubc6 and the sybody were present in the cryo-EM sample, no extra densities corresponding to them were observed. This is unfortunate since understanding the association of Ubc6 with Doa10 could provide significant insights into how Doa10 facilitates the transfer of Ub from Ubc6 to its substrates. Given that Ubc6 and Doa10 were co-eluted during gel filtration, it is plausible that they are bound even on the grid. Supplementary Fig. 2 lacks 3D classification, which is often necessary to obtain additional maps for weakly-bound or conformationally flexible proteins. Therefore, it is recommended to perform extensive 3D classification to potentially visualize the Ubc6 map and improve the map resolution.

We agree with the reviewer's point that it is unfortunate that we could not observe clear densities corresponding to Ubc6 in our cryo-EM reconstruction. Presumably, this is due to the dynamic nature of E3-E2 interactions, which in our experience has often required substantial protein engineering and/or chemical stabilization prior to successful structure determination. In addition, Ubc6 is also reported to be a substrate of Doa10, which might create further heterogeneity, for example through multiple binding modes for Ubc6 on the ligase, confounding interpretation of binding assays targeting only one potential binding site.

As requested by the reviewer, we have included more detailed 3D classification figures in Supplementary Figure 2b, which illustrate that even with extensive classification, subclasses with resolved Ubc6 could not be obtained.

We have modified the relevant parts in the manuscript to reflect these changes:

“The cryo-EM map is dominated by a well-resolved, large domain of nanodisc-enwrapped transmembrane helices (Supplementary Fig. 3a). Although our cryo-EM sample contained Ubc6 and the sybody, we did not observe unambiguous density for either, even after extensive 3D classification (Supplementary Fig. 2b).”

3. While the molecular architecture of Doa10 itself provides valuable information, several important hypotheses put forth by the authors rely on the docking model by AF multimer (Fig. 4). Although the mutational data is appreciated, it is hypothesized that MARCH6, Ube2J2, and Ub bind at the exact binding interface as suggested by the docking model. Thus, it is crucial to conduct further binding experiments using recombinant proteins to validate the proposed model.

In response to this comment and a related suggestion from Reviewer 1, we have now included a ubiquitin discharge assay, which probes mutations at the E2-ubiquitin and RING-E2 interfaces, aimed at validating the AF multimer predicted, ternary model of the MARCH6 RING domain with the Ube2J2 catalytic domain and ubiquitin (new Figure 4b, c and Supplementary Fig. 8a-d). The discharge assay is an established functional assay to verify the formation of a ternary RING-E2~ubiquitin complex, in which the RING domain stabilizes the closed E2~ubiquitin conformation to promote discharge of the ubiquitin from the E2~ubiquitin complex (Saha et al., 2011, Mol. Cell.).

We first mutated the predicted interface between the MARCH6 RING and the Ube2J2 catalytic domain (Supplementary Fig. 8a), which resembles previously observed RING-E2 interactions, as exemplified by the RING-UBCH5B~ubiquitin structures (Plechanovová et al., 2012, Nature; Dou et al., 2012, Nat. Struct. Mol. Biol.). Here, mutating the conserved MARCH6 V11 to either V11A or V11D caused moderate to severe discharge defects, respectively (Fig. 4b, Supplementary Fig. 8b), consistent with the hydrophobic interactions between these domains.

Next, we mutated the predicted Ube2J2-ubiquitin interface, centered on the ubiquitin I44 patch (Supplementary Fig. 8a). Here, mutating either Ube2J2 S120 or T116 to bulky residues caused strong discharge defects, which could be partially rescued when using a reciprocal ubiquitin I44A mutant (Fig. 4c, Supplementary Fig. 8c). Mutation of these two Ube2J2 residues to a smaller glycine resulted in discharge defects which could not be rescued by ubiquitin I44A (Supplementary Fig. 8c).

Together, our mutational analysis of the RING-E2 and E2-ubiquitin interfaces support the AF model for a ternary MARCH6(RING)-Ube2J2-ubiquitin catalytic complex.

Furthermore, we used this discharge assay to test the MARCH6^{E5A/E6A/E15A/E19A} mutation, which caused a strong defect in our SQLE stability assay. We suggested a catalytic role for this mutant based on the close proximity to the RING domain, the AF model of the catalytic moieties and the increase in protein abundance (similar to other catalytic defective mutants like Δ RING). Indeed, the mutation also results in a strong discharge defect (Fig. 4b) further supporting a possible role in stabilizing the catalytic domains for ubiquitin discharge.

We have rewritten the relevant part of the Results section to incorporate these new findings:

“For structural insights into mutants suggested to play a catalytic role, we used AF-multimer⁵² to model a MARCH6-ubiquitin-Ube2J2 complex for both the full-length protein complex, spanning the membrane (Fig. 4a), and only the soluble catalytic domains focusing on the ubiquitination active site (Supplementary Fig. 8a). Despite close resemblance to known RING-E2~ubiquitin structures (Supplementary Fig. 8a), we tested the predicted positioning of the catalytic moieties using an ubiquitin discharge assay^{53, 54}. This assay follows the release of ubiquitin from the thioester bond with Ube2J2, stimulated by interactions between E2~ubiquitin and the RING domain. Indeed, adding WT

MARCH6 RING to the loaded Ube2J2~ubiquitin complex accelerates ubiquitin discharge in our in vitro assay (Fig. 4b and Supplementary Fig. 8b).

To further assess the positioning of the catalytic domains, we mutated conserved residues predicted to be localized at the interfaces between these domains (Supplementary Fig. 8a). First, we mutated MARCH6 RING V11 predicted to interact with a hydrophobic patch in UBE2J2 (Supplementary Fig. 8a). Consistent with the model, mutations to aspartic acid or alanine lead to a strong or intermediate defect, respectively, in RING-mediated ubiquitin discharge from Ube2J2~ubiquitin (Fig. 4b, Supplementary Fig. 8b). Next, we targeted the predicted interface between the E2 catalytic domain with the ubiquitin I44 patch (Fig. 4c, Supplementary Fig. 8a, c, d). Introducing bulky residues at this interface (S120F, S120R, T116F, T116R) results in impaired ubiquitin discharge with WT ubiquitin. These discharge defects are partially rescued with a ubiquitin I44A mutation designed to compensate for the larger opposing E2 residues (Fig. 4c, Supplementary Fig. 8c). Furthermore, E2 mutations to glycine have milder effects, which are not rescued by ubiquitin I44A in agreement with the model (Supplementary Fig. 8c).

In our broad mutagenesis campaign, we found mutations in the MARCH6 N-terminus (MARCH6^{E5A/E6A/E15A/E19A}) highly defective in mediating SQLE degradation (Fig. 3b, cyan). Our MARCH6-Ube2J2-ubiquitin AF model predicts MARCH6 E5 to form a bond with Ube2J2 residue K26 (Fig. 4d), suggesting additional catalytic domain stabilization by this N-terminus. Indeed, the isolated MARCH6^{E5A/E6A/E15A/E19A} RING domain has a defect in stimulating ubiquitin discharge from Ube2J2 compared to WT (Fig. 4b), supporting a role of the N-terminus. Finally, we also observed accumulation of MARCH6^{E5A/E6A/E15A/E19A} protein, emphasizing the importance of the MARCH6 N-terminus for catalysis, including auto-degradation (Supplementary Fig. 6a)."

4. The models described in the second and third paragraphs of the discussion are intriguing; however, they remain highly speculative. To persuade readers, it is necessary to provide supporting experimental results from either the authors or other relevant literature.

Our revised manuscript contains new data, including 1) thorough mutational data targeting AF predicted RING-E2 and E2-ubiquitin interfaces, functionally probed using an established E2~ubiquitin discharge assay (Fig. 4b, c) and 2), and IP-WB data to show weakened interaction between MARCH6 and SQLE in a subset of membrane channel lining residues (Fig. 5d). Together with the AF multimer predictions, these new data support the models we discuss in the second and third paragraphs of the Discussion.

We have nevertheless made some adjustments to these paragraphs in order to make them less speculative, as pointed out by the reviewer.

We have modified the relevant parts in the Discussion reflect these changes:

"Precise positioning of the RING domain above the central channel seems crucial for ubiquitylation, as seen in the cryo-EM model of Doa10, the AF predicted ternary E3-E2-ubiquitin complex and mutational probing of this ternary model. Unlike many E3 ligases^{9, 58}, the Doa10/MARCH6 RING domain is not flexibly tethered to the remainder of the structure. Instead, it is positioned by a conserved β -sheet formed by the residues downstream of the RING domain and the C-terminus. Our structure-wide screening across MARCH6, as well as previous targeted mutagenesis of Doa10^{3, 43}, showed the importance of this pillar-like structure. Moreover, one side of the β -sheet connects via Helix 1 to the lipid-bound scaffold, and the other side connects to Helix 15 in the flexible gate. These functionally-important regions are predicted to be involved in substrate engagement. Thus, the overall Doa10/MARCH6 architecture interconnects the key functional elements (Fig. 7a).

In addition, the conformational flexibility of parts of the Doa10/MARCH6 membrane domain, as observed by cryo-EM and AF prediction, could enable Doa10/MARCH6 regulation of diverse substrates. It seems that two key factors would govern the efficiency of substrate engagement for ubiquitylation: 1) the modulation of membrane channel dimensions, and potentially substrate access, via the flexibly-tethered gate, and 2) specific properties of the membrane channel lining. Also, given the diversity of ERAD substrates, it seems that there could be more than one substrate-binding modality, for example if the gate helices themselves could fold into the central channel to create a neo substrate-binding surface. Meanwhile, it seems that the intricate connections to the RING domain would allow for subtle reorientation of the ubiquitylation active site in response to substrate and lipid binding, for both modification of diverse substrates and regulation in response to lipid interactions.”

5. The interpretation of lipid-binding sites within the scaffold domain is based on mutational results of residues coordinated with lipids. The authors speculate that cholesterol might bind to these specific sites due to MARCH6's role in regulating cholesterol homeostasis. However, this speculation is not sufficiently supported. Cholesterol density can usually be distinguished from that of phospholipids, and intercalated phospholipids often stabilize the overall architecture of membrane proteins. Therefore, the hypothesis regarding cholesterol binding requires more evidence.

We agree with the reviewer. In our revised manuscript we took care to emphasize that while we could model four tightly bound phospholipids in our Doa10 structure – and find conserved sites in human MARCH6 which we functionally test using our reporter assay – we cannot make any claims as to whether or not these sites could also bind sterols. As illustrated in Figure 2e, we see a plethora of non-protein densities at the periphery of our Doa10 model, but the resolution of the map is not sufficient to allow non-ambiguous modeling of sterol molecules into these peripheral densities.

We have modified the Discussion in response to the Reviewer's suggestion as follows:

“Notably, MARCH6 regulates cholesterol homeostasis, a process which is on many levels tightly controlled by cholesterol or other lipids^{24, 59, 60, 61, 62, 63, 64, 65, 66}. Indeed, cholesterol potentiates degradation of the SQLE reporter in our flow-cytometry screen. Compellingly, we also found well-resolved phospholipids interacting with the scaffold in our Doa10 structure. Our finding that mutation of these sites stabilizes SQLE, without diminishing MARCH6 protein levels, raises the exciting possibility that substrate regulation might be tied to sensing of metabolic signals within the lipid bilayer via MARCH6 conformation.”

6. Considering the relatively large deviation of the median mCherry:GFP ratios for each MARCH6 variant, the number of repeats is insufficient, especially for critical mutants that support the main results of the manuscript. It is recommended to increase the number of repeats for critical mutants and provide p-values.

In response to the Reviewer, we have now performed additional replicates of our SQLE reporter assay (at least four independent biological replicates for each mutant). In response to this comment and a related comment from Reviewer 2, we now also include error bars (S.E.M.) and ANOVA analysis for each mutant relative to the WT control, as well as showing the individual data points in our bar graph. These updates are shown in Figure 3, with the results of the ANOVA provided for each mutant in Supplementary Table 2. A total of 24 mutants out of the 95 tested variants show significant changes compared to WT.

Minor comments:

1. The observation that extra densities are distinct depending on the size of nanodiscs (Fig. 2C) is interesting. It suggests that the flexibility of additional maps can be modulated by nanodisc size. It

would be worthwhile to explore other MSP proteins with a wider range of size distributions to visualize Ubc6.

In our manuscript, we provide Doa10 cryo-EM maps reconstituted in two different nanodisc sizes (MSP1E3D1, \varnothing ~12.9 nm and MSP2N2, \varnothing 15-16.5 nm). Although we tried narrower disc sizes (MSP1D1, \varnothing ~9.7 nm), we could no longer achieve quality cryo-EM reconstructions.

2. Conducting 3D variability analysis would be useful to assess the structural flexibility of the gate (Fig. 7).

We did attempt to resolve additional continuous flexibility and/or discrete heterogeneity in our high-resolution Doa10 cryo-EM dataset using 3D variability analysis. However, this did not prove to be informative. The flexibility of the gate could be largely constrained by the MSP1E3D1 nanodisc and thus difficult to capture by 3D variability analysis.

3. The discussion section should be thoroughly rewritten to minimize speculation and consider alternative explanations supported by previous findings in the literature.

We have taken care to edit our Discussion to more closely reflect the data available to us and contextualize our proposed models to what has been previously published in the literature. This is reflected in the modified Discussion paragraphs we have provided above to Reviewer #3 Point 4.

REVIEWERS' COMMENTS

Reviewer #1 (Remarks to the Author):

All my concerns have been addressed and this manuscript is suitable for publication in this Reviewer's opinion.

Reviewer #3 (Remarks to the Author):

The manuscript has been revised and significantly improved by the authors. The revisions and descriptions are fair, careful, and grounded in an objective point of view. The authors have appropriately addressed all of my concerns. I recommend this manuscript for publication.